# ReQFlow: Rectified Quaternion Flow for Efficient and High-Quality Protein Backbone Generation

Angxiao Yue [1]   Zichong Wang [2]   Hongteng Xu [1 3 4]

## Abstract

Protein backbone generation plays a central role in *de novo* protein design and is significant for many biological and medical applications. Although diffusion and flow-based generative models provide potential solutions to this challenging task, they often generate proteins with undesired designability and suffer computational inefficiency. In this study, we propose a novel rectified quaternion flow (ReQFlow) matching method for fast and high-quality protein backbone generation. In particular, our method generates a local translation and a 3D rotation from random noise for each residue in a protein chain, which represents each 3D rotation as a unit quaternion and constructs its flow by spherical linear interpolation (SLERP) in an exponential format. We train the model by quaternion flow (QFlow) matching with guaranteed numerical stability and rectify the QFlow model to accelerate its inference and improve the designability of generated protein backbones, leading to the proposed ReQFlow model. Experiments show that ReQFlow achieves on-par performance in protein backbone generation while requiring much fewer sampling steps and significantly less inference time (e.g., being $37\times$ faster than RFDiffusion and $63\times$ faster than Genie2 when generating a backbone of length 300), demonstrating its effectiveness and efficiency. Code is available at https://github.com/AngxiaoYue/ReQFlow.

## 1. Introduction

*De novo* protein design (Ingraham et al., 2023; Lin & Alquraishi, 2023) aims to design rational proteins from scratch with specific properties or functions, which has many biological and medical applications, such as developing novel enzymes for biocatalysis (Kelly et al., 2020) and discovering new drugs for diseases (Teague, 2003; Silva et al., 2019). This task is challenging due to the extremely huge design space of proteins. For simplifying the task, the mainstream *de novo* protein design strategy takes protein backbone generation (i.e., generating 3D protein structures without side chains) as the key step that largely determines the rationality and basic properties of designed proteins.

Focusing on protein backbone generation, many deep generative models, especially those diffusion and flow-based models (Ho et al., 2020; Watson et al., 2023; Lin et al., 2024; Yim et al., 2023b;a; Bose et al., 2024; Huguet et al., 2024; Lipman et al., 2023), have been proposed as potential solutions. However, these models often generate protein backbones with poor designability (the key metric indicating the quality of generated protein backbones), especially for proteins with long residue chains. In addition, diffusion or flow models often require many sampling steps to generate protein backbones, resulting in high computational complexity and long inference time. As a result, the above drawbacks on generation quality and computational efficiency limit these models in practical large-scale applications.

To overcome the above challenges, we propose a novel rectified quaternion flow (ReQFlow) matching method, achieving fast and high-quality protein backbone generation. As illustrated in Figure 1a, our method learns a model to generate a local 3D translation and a 3D rotation respectively from random noise for each residue in a protein chain. Different from existing models, the proposed model represents each rotation as a unit quaternion and constructs its quaternion flow in SO(3) by spherical linear interpolation (SLERP) in an exponential format (Sola, 2017), which can be learned by our quaternion flow (QFlow) matching strategy. Furthermore, given a trained QFlow model, we leverage the rectified flow technique in (Liu, 2022), re-training the model based on the paired noise and protein backbones generated by the model itself. The rectified QFlow (i.e., ReQFlow)

---

[1]Gaoling School of Artificial Intelligence, Renmin University of China, Beijing, China [2]School of Statistics, Renmin University of China, Beijing, China [3]Beijing Key Laboratory of Research on Large Models and Intelligent Governance [4]Engineering Research Center of Next-Generation Intelligent Search and Recommendation, MOE. Correspondence to: Hongteng Xu <hongtengxu@ruc.edu.cn>.

*Proceedings of the 42nd International Conference on Machine Learning*, Vancouver, Canada. PMLR 267, 2025. Copyright 2025 by the author(s).

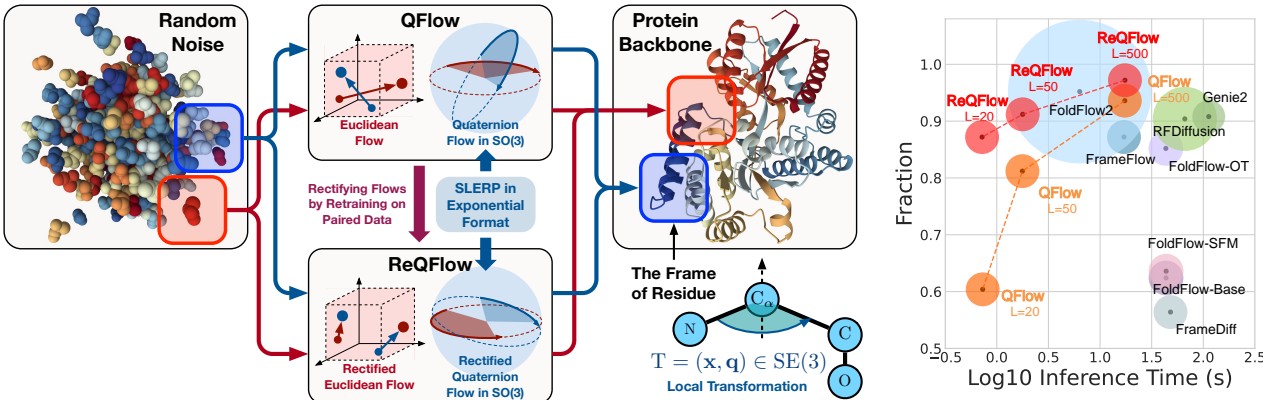

(a) The scheme of rectified quaternion flow matching     (b) Comparisons for various methods

*Figure 1.* (a) An illustration of our rectified quaternion flow matching method, in which each residue is represented as a frame associated with a local transformation. (b) For each method, the size of its circle indicates the model size, and the location of the circle's centroid indicates the logarithm of the average inference time when generating a protein backbone with length $N = 300$ and the Fraction score of designable protein backbones. For QFlow and ReQFlow, we set the sampling step $L \in \{20, 50, 500\}$, respectively.

model leads to straightened sampling paths in $\mathbb{R}^3$ and SO(3), respectively, when generating translations and rotations. As a result, we can apply fewer sampling steps to accelerate the generation process significantly.

We demonstrate the rationality and effectiveness of Re-QFlow compared to existing diffusion and flow-based methods. In particular, thanks to the exponential format SLERP, ReQFlow is learned and implemented with guaranteed numerical stability and computational efficiency, especially when the rotation angle is close to 0 or $\pi$. Experimental results demonstrate that ReQFlow achieves competitive performance, generating high-quality protein backbones with significantly reduced inference time. Furthermore, Re-QFlow consistently maintains effectiveness and efficiency in generating long-chain protein backbones (e.g., the protein backbones with over 500 residues), where all baseline models suffer severe performance degradation. As shown in Figure 1b, ReQFlow outperforms existing methods and generates high-quality protein backbones, whose designability Fraction score is 0.972 when sampling 500 steps and 0.912 when merely sampling 50 steps.

## 2. Related Work and Preliminaries

### 2.1. Protein Backbone Generation

Many diffusion and flow-based methods have been proposed to generate protein backbones. These methods often parameterize protein backbones like AlphaFold2 (Jumper et al., 2021) does, representing each protein's residues as a set of SE(3) frames (Yim et al., 2023b;a; Wang et al., 2025). Accordingly, FrameDiff (Yim et al., 2023b) generates protein backbones by two independent diffusion processes, generating the corresponding frames' local translations and ro-

tations, respectively. Following the same framework, flow-based methods like FrameFlow (Yim et al., 2023a) and FoldFlow (Bose et al., 2024) replace the stochastic diffusion processes with deterministic flows.

For the above methods, many efforts have been made to modify their model architectures and improve data representations, e.g., the Clifford frame attention module in GAFL (Wagner et al., 2024) and the asymmetric protein representation module in Genie (Lin & Alquraishi, 2023) and Genie2 (Lin et al., 2024). In addition, some methods leverage large-scale pre-trained models to improve generation quality. For example, RFDiffusion (Watson et al., 2023) utilizes the pre-trained RoseTTAFold (Baek et al., 2021) as the backbone model. FoldFlow2 (Huguet et al., 2024) improves FoldFlow by using a protein large language model for residue sequence encoding. Taking scaling further and adopting a different architectural approach, Proteína (Geffner et al., 2025) developed a large-scale, flow-based generative model using a non-equivariant transformer operating directly on C-alpha coordinates.

Currently, the above methods often suffer the conflict on computational efficiency and generation quality. The state-of-the-art methods like RFDiffusion (Watson et al., 2023) and Genie2 (Lin et al., 2024) need long inference time to generate protein backbones with reasonable quality. Frame-Flow (Yim et al., 2023a) and GAFL (Wagner et al., 2024) significantly improve inference speed while lag behind RFD-iffusion and Genie2 in protein backbone quality. Moreover, these methods suffer severe performance degradation when generating long-chain protein backbones. These limitations motivate us to develop the proposed ReQFlow, improving the current flow-based methods and generating protein backbones efficiently with satisfactory designability.

## 2.2. Quaternion Algebra and Its Applications

The proposed ReQFlow is designed based on quaternion algebra (Dam et al., 1998; Zhu et al., 2018). Mathematically, quaternion is an extension of complex numbers into four-dimensional space, consists of one real component and three orthogonal imaginary components. A quaternion is formally expressed as $q = s + x\mathtt{i} + y\mathtt{j} + z\mathtt{k} \in \mathbb{H}$, where $\mathbb{H}$ denotes the quaternion domain, and $s, x, y, z \in \mathbb{R}$. The imaginary components $\{\mathtt{i}, \mathtt{j}, \mathtt{k}\}$ satisfy $\mathtt{i}^2 = \mathtt{j}^2 = \mathtt{k}^2 = \mathtt{ijk} = -1$. Each $q \in \mathbb{H}$ can be equivalently represented as a vector $\boldsymbol{q} = [s, \boldsymbol{u}^\top]^\top \in \mathbb{R}^4$, where $\boldsymbol{u}^\top = [x, y, z]^\top$. Given $\boldsymbol{q}_1 = [s_1, \boldsymbol{u}_1^\top]^\top$ and $\boldsymbol{q}_2 = [s_2, \boldsymbol{u}_2^\top]^\top$, their multiplication is achieved by Hamilton product, i.e.,

$$\boldsymbol{q}_1 \otimes \boldsymbol{q}_2 = \begin{bmatrix} s_1 s_2 - \boldsymbol{u}_1^\top \boldsymbol{u}_2 \\ s_1 \boldsymbol{u}_2 + s_2 \boldsymbol{u}_1 + \boldsymbol{u}_1 \times \boldsymbol{u}_2 \end{bmatrix}, \qquad (1)$$

where $\times$ denotes the cross product.

Quaternion is a powerful tool to describe 3D rotations. For a 3D rotation in the axis-angle formulation, i.e., $\boldsymbol{\omega} = \phi\boldsymbol{u} \in \mathbb{R}^3$, where the unit vector $\boldsymbol{u}$ and the scalar $\phi$ denote the rotation axis and angle, respectively, we can convert it to a unit quaternion by an exponential map (Sola, 2017):

$$\boldsymbol{q} = \exp\left(\frac{1}{2}\boldsymbol{\omega}\right) = \left[\cos\frac{\phi}{2}, \sin\frac{\phi}{2}\boldsymbol{u}^\top\right]^\top \in \mathbb{S}^3, \quad (2)$$

where $\mathbb{S}^3 = \{\boldsymbol{q} \in \mathbb{R}^4 \mid \|\boldsymbol{q}\|_2 = 1\}$ is the 4D hypersphere. The conversion from a unit quaternion to an angle-axis representation is achieved by a logarithmic map:

$$\boldsymbol{\omega} = 2\log(\boldsymbol{q}). \qquad (3)$$

Suppose that we rotate a point $\boldsymbol{v}_1 \in \mathbb{R}^3$ to $\boldsymbol{v}_2$ by $\boldsymbol{\omega}$, we can equivalently implement the operation by

$$\boldsymbol{v}_2 = \text{Im}(\boldsymbol{q} \otimes [0, \boldsymbol{v}_1^\top]^\top \otimes \boldsymbol{q}^{-1}), \qquad (4)$$

where $\boldsymbol{q}^{-1} = [\cos\frac{\phi}{2}, -\sin\frac{\phi}{2}\boldsymbol{u}^\top]^\top$ is the inverse of $\boldsymbol{q}$ and "Im($\cdot$)" denotes the imaginary components of a quaternion (i.e., the last three elements of the corresponding 4D vector). The quaternion-based rotation representation in Eq. (4) offers several advantages, including compactness, computational efficiency, and avoidance of gimbal lock (Hemingway & O'Reilly, 2018), which has been widely used in skeletal animation (Shoemake, 1985), robotics (Pervin & Webb, 1982), and virtual reality (Kuipers, 1999).

Besides computer graphics, some quaternion-based machine learning models have been proposed for other tasks, e.g., image processing (Xu et al., 2015; Zhu et al., 2018) and structured data (e.g., graphs and point clouds) analysis (Zhang et al., 2020; Zhao et al., 2020). Recently, some quaternion-based models have been developed for scientific problems, e.g., the quaternion message passing (Yue et al., 2024) for molecular conformation representation (Gasteiger

et al., 2020; Wang et al., 2023) and the quaternion generative models for molecule generation (Köhler et al., 2023; Guo et al., 2025). However, the computational quaternion techniques are seldom considered in protein-related tasks. Our work fill this blank, demonstrating the usefulness of quaternion algebra in protein backbone generation.

## 3. Proposed Method

### 3.1. Protein Backbone Parameterization

We parameterize the protein backbone following (Jumper et al., 2021; Yim et al., 2023b;a; Bose et al., 2024). As illustrated in Figure 1a, each residue is represented as a frame, where the frame encodes a rigid transformation starting from the idealized coordinates of four heavy atoms: $[\text{N}^*, \text{C}_\alpha^*, \text{C}^*, \text{O}^*] \in \mathbb{R}^{3\times4}$. In this representation, $\text{C}_\alpha^* = [0, 0, 0]^\top$ is placed at the origin, and the transformation incorporates experimental bond angles and lengths (Engh & Huber, 2012). We can derive each residue's frame by

$$[\text{N}^i, \text{C}_\alpha^i, \text{C}^i, \text{O}^i] = \text{T}^i \circ [\text{N}^*, \text{C}_\alpha^*, \text{C}^*, \text{O}^*], \qquad (5)$$

where $\text{T}^i \in \text{SE}(3)$ is the local orientation-preserving rigid transformation mapping the idealized frame to the frame of the $i$-th residue. In this study, we represent $\text{T}^i = (\boldsymbol{x}^i, \boldsymbol{q}^i)$, where $\boldsymbol{x}^i \in \mathbb{R}^3$ represents the 3D translation and a unit quaternion $\boldsymbol{q}^i \in \mathbb{S}^3$, which double-covers SO(3), represents a 3D rotation. According to Eq. (4), the action of $\text{T}^i$ on a coordinate $\boldsymbol{v} \in \mathbb{R}^3$ can be implemented as

$$\text{T}^i \circ \boldsymbol{v} = \boldsymbol{x}^i + \text{Im}(\boldsymbol{q} \otimes [0, \boldsymbol{v}^\top]^\top \otimes \boldsymbol{q}^{-1}). \qquad (6)$$

Note that, for protein backbone generation, we can use the planar geometry of backbone to impute the coordinate of the oxygen atom $\text{O}^i$ (Yim et al., 2023a; Watson et al., 2023), so we do not need to parameterize the rotation angle of the bond "$\text{C}_\alpha - \text{C}$". As a result, for a protein backbone with $N$ residues, we have a collection of $N$ frames, resulting in the parametrization set $\Theta = \{\text{T}^i\}_{i=1}^N$. Therefore, we can formulate the protein backbone generation problem as modeling and generating $\{\text{T}^i\}_{i=1}^N$ automatically.

### 3.2. Quaternion Flow Matching

We decouple the translation and rotation of each frame, establishing two independent flows in $\mathbb{R}^3$ and SO(3), respectively. Without the loss of generality, we define these two flows in the time interval $[0, 1]$. When $t = 0$, we sample the starting points of the flows as random noise, i.e., $\text{T}_0 = (\boldsymbol{x}_0, \boldsymbol{q}_0) \sim \mathcal{T}_0 \times \mathcal{Q}_0$, where $\mathcal{T}_0 = \mathcal{N}(\boldsymbol{0}, \mathbf{I}_3)$ is the Gaussian distribution for translations, and $\mathcal{Q}_0 = \mathcal{IG}_{\text{SO}(3)}$ is the isotropic Gaussian distribution on SO(3) for rotations (Leach et al., 2022), corresponding to uniformly sampling rotation axis $\boldsymbol{u} \in \mathbb{S}^2$ and rotation angle $\phi \in [0, \pi]$

with the density:

$$f(\phi) = \frac{1 - \cos \phi}{\pi} \sum_{l=0}^{\infty} (2l+1) e^{-l(l+1)\epsilon^2} \frac{\sin((l + \frac{1}{2})\phi)}{\sin(\phi/2)}.$$

Based on Eq. (2), we convert the sampled axis and angle to $\boldsymbol{q}_0$. When $t = 1$, the ending points of these two flows, denoted as $\mathrm{T}_1 = (\boldsymbol{x}_1, \boldsymbol{q}_1)$, should be the transformation of a frame. We denote the data distribution of $\mathrm{T}_1$ as $\mathcal{T}_1 \times \mathcal{Q}_1$.

**Linear Interpolation of Translation.** For $\boldsymbol{x}_0 \sim \mathcal{T}_0$ and $\boldsymbol{x}_1 \sim \mathcal{T}_1$, we can interpolate the trajectory between them linearly: for $t \in [0, 1]$,

$$\boldsymbol{x}_t = (1-t)\boldsymbol{x}_0 + t\boldsymbol{x}_1,$$
with constant translation velocity: $\boldsymbol{v} = \boldsymbol{x}_1 - \boldsymbol{x}_0$. $\quad$ (7)

**SLERP of Rotation in Exponential Format.** For unit quaternions $\boldsymbol{q}_0 \sim \mathcal{Q}_0$ and $\boldsymbol{q}_1 \sim \mathcal{Q}_1$, we interpolate the trajectory between them via SLERP in an exponential format (Sola, 2017):

$$\boldsymbol{q}_t = \boldsymbol{q}_0 \otimes \exp(t \log(\boldsymbol{q}_0^{-1} \otimes \boldsymbol{q}_1)),$$
with constant angular velocity: $\boldsymbol{\omega} = \phi \boldsymbol{u}$. $\quad$ (8)

Here, $\boldsymbol{q}_0^{-1} \otimes \boldsymbol{q}_1 = [\cos(\phi/2), \sin(\phi/2) \boldsymbol{u}^\top]^\top$ and $\boldsymbol{\omega} = 2\log(\boldsymbol{q}_0^{-1} \otimes \boldsymbol{q}_1)$. $\exp(\cdot)$ and $\log(\cdot)$ are exponential and logarithmic maps defined in Eq. (2) and Eq. (3), respectively.

**Training QFlow Model.** In this study, we adopt the SE(3)-equivariant neural network in FrameFlow (Yim et al., 2023a), denoted as $\mathcal{M}_\theta$, to model the flows. Given the transformation at time $t$, i.e., $\mathrm{T}_t$, the model predicts the transformation at $t = 1$:

$$\mathrm{T}_{\theta,1} = (\boldsymbol{x}_{\theta,1}, \boldsymbol{q}_{\theta,1}) = \mathcal{M}_\theta(\mathrm{T}_t, t). \quad (9)$$

We train this model by the proposed quaternion flow (QFlow) matching method. In particular, given the frame $\mathrm{T}_1 = (\boldsymbol{x}_1, \boldsymbol{q}_1)$, we first sample a timestamp $t \sim \mathrm{Uniform}([0,1])$ and random initial points $\mathrm{T}_0 = (\boldsymbol{x}_0, \boldsymbol{q}_0) \sim \mathcal{T}_0 \times \mathcal{Q}_0$. Then, we derive $(\boldsymbol{x}_t, \boldsymbol{v})$ and $(\boldsymbol{q}_t, \boldsymbol{\omega})$ via Eq. (7) and Eq. (8), respectively. Passing $(\boldsymbol{x}_t, \boldsymbol{q}_t, t)$ through the model $\mathcal{M}_\theta$, we obtain $\boldsymbol{x}_{\theta,1}$ and $\boldsymbol{q}_{\theta,1}$, and derive the translation and angular velocities at time $t$ by

$$\boldsymbol{v}_{\theta,t} = \frac{\boldsymbol{x}_{\theta,1} - \boldsymbol{x}_t}{1-t}, \quad \boldsymbol{\omega}_{\theta,t} = \frac{2\log(\boldsymbol{q}_t^{-1} \otimes \boldsymbol{q}_{\theta,1})}{1-t}. \quad (10)$$

Based on the constancy of the velocities, we train the model $\mathcal{M}_\theta$ by minimizing the following two objectives:

$$\begin{aligned} \mathcal{L}_{\mathbb{R}^3} &= \mathbb{E}_{t,\mathcal{T}_0,\mathcal{T}_1}[\|\boldsymbol{v} - \boldsymbol{v}_{\theta,t}\|^2], \\ \mathcal{L}_{\mathrm{SO}(3)} &= \mathbb{E}_{t,\mathcal{Q}_0,\mathcal{Q}_1}[\|\boldsymbol{\omega} - \boldsymbol{\omega}_{\theta,t}\|^2]. \end{aligned} \quad (11)$$

Besides the above MSE losses, we further consider the auxiliary loss proposed in (Yim et al., 2023b), which discourages physical violations, e.g., chain breaks or steric clashes. Therefore, we train the model by

$$\min_\theta \mathcal{L}_{\mathbb{R}^3} + \mathcal{L}_{\mathrm{SO}(3)} + \alpha \cdot \mathbf{1}\{t < \zeta\} \cdot \mathcal{L}_{\mathrm{aux}}, \quad (12)$$

where $\alpha \geq 0$ is the weight of the auxiliary loss, $\mathbf{1}$ is an indicator, signifying that the auxiliary loss is applied only when $t$ is sampled below a predefined threshold $\zeta$.

**Inference Based on QFlow.** Given a trained model, we can generate frames of residues from noise with the predicted velocities. In particular, given initial $(\boldsymbol{x}_0, \boldsymbol{q}_0) \sim \mathcal{T}_0 \times \mathcal{Q}_0$, the translation is generated by an Euler solver with $L$ steps:

$$\boldsymbol{x}_{t+\Delta t} = \boldsymbol{x}_t + \boldsymbol{v}_{\theta,t} \cdot \Delta t, \quad (13)$$

where the step size $\Delta t = \frac{1}{L}$. The quaternion of rotation is generated with an exponential step size scheduler: We modify Eq. (8), interpolating $\boldsymbol{q}_t$ with an acceleration as

$$\boldsymbol{q}_t = \boldsymbol{q}_0 \otimes \exp((1 - e^{-\gamma t}) \log(\boldsymbol{q}_0^{-1} \otimes \boldsymbol{q}_1)), \quad (14)$$

where $\gamma$ controls the rotation accelerating, and we empirically set $\gamma = 10$. Then, the Euler solver becomes:

$$\boldsymbol{q}_{t+\Delta t} = \boldsymbol{q}_t \otimes \exp\left(\frac{1}{2}\Delta t \cdot \gamma e^{-\gamma t} \boldsymbol{\omega}_{\theta,t}\right), \quad (15)$$

where $\gamma e^{-\gamma t} \boldsymbol{\omega}_{\theta,t}$ is the adjusted angular velocity. Previous works (Bose et al., 2024; Yim et al., 2023a) have demonstrated that the exponential step size scheduler helps reduce sampling steps and enhance model performance.

### 3.3. Rectified Quaternion Flow

Given the trained QFlow model $\mathcal{M}_\theta$, we can rectify the flows in $\mathbb{R}^3$ and SO(3), respectively, by applying the flow rectification method (Liu, 2022). In particular, we generate noisy $\mathrm{T}_0' = \{\boldsymbol{x}_0', \boldsymbol{q}_0'\} \sim \mathcal{T}_0 \times \mathcal{Q}_0$ and transfer to $\mathrm{T}_1' = \{\boldsymbol{x}_1', \boldsymbol{q}_1'\} \sim \mathcal{T}_1 \times \mathcal{Q}_1$ by $\mathcal{M}_\theta$. Taking $\mathcal{M}_\theta$ as the initialization, we use the noise-sample pairs, i.e., $\{\mathrm{T}_0', \mathrm{T}_1'\}$, to train the model further by the same loss in Eq. (12) and derive the rectified QFlow (ReQFlow) model.

The work in (Liu, 2022) has demonstrated that the rectified flow of translation in $\mathbb{R}^3$ preserves the marginal law of the original translation flow and reduces the transport cost from the noise to the samples. We find that these theoretical properties are also held by the rectified quaternion flow under mild assumptions. Let $(\boldsymbol{q}_0, \boldsymbol{q}_1) \sim \mathcal{Q}_0 \times \mathcal{Q}_1$ be the pair used to train QFlow, and $(\boldsymbol{q}_0', \boldsymbol{q}_1')$ be the pair induced from $(\boldsymbol{q}_0, \boldsymbol{q}_1)$ by flow rectification. Then, we have

**Theorem 3.1.** *(**Marginal preserving property**). The pair $(\boldsymbol{q}_0', \boldsymbol{q}_1')$ is a coupling of $\mathcal{Q}_0$ and $\mathcal{Q}_1$. The marginal law of $\boldsymbol{q}_t'$ equals that of $\boldsymbol{q}_t$ at everytime, that is $Law(\boldsymbol{q}_t') = Law(\boldsymbol{q}_t)$.*

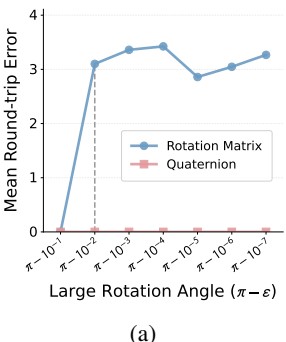

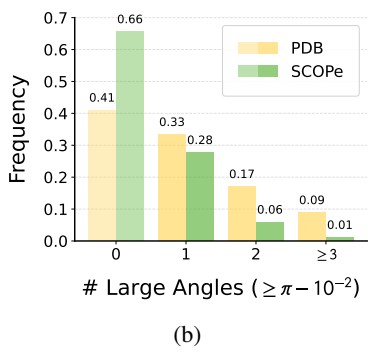

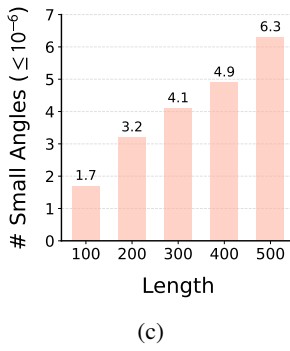

| (a) | (b) | (c) |
|---|---|---|

*Figure 2.* (a) Mean round-trip errors from $\pi - 10^{-1}$ to $\pi - 10^{-7}$. (b) The frequency of suffering large rotation angles per protein when training on the two datasets. (c) The average number of small rotation angles per protein when generating ten backbones for each length.

*Table 1.* Comparisons for various rotation interpolation methods.

| Method | | Matrix Geodesic | SLERP (Add. Format) | SLERP (Exp. Format) |
|---|---|---|---|---|
| **Interpolation** | Formula | $\boldsymbol{R}_0 \exp_M \left( t \log_M(\boldsymbol{R}_0^\top \boldsymbol{R}_1) \right)$ | $\frac{\sin((1-t)\frac{\phi}{2})}{\sin\frac{\phi}{2}} \boldsymbol{q}_0 + \frac{\sin(t\frac{\phi}{2})}{\sin\frac{\phi}{2}} \boldsymbol{q}_1$ | $\boldsymbol{q}_0 \otimes \exp(t \log(\boldsymbol{q}_0^{-1} \otimes \boldsymbol{q}_1))$ |
| | Velocity | $\boldsymbol{\Omega} = \log_M(\boldsymbol{R}_0^\top \boldsymbol{R}_1)$ | $\boldsymbol{\eta}_t = \frac{\phi(\cos(\frac{t\phi}{2})\boldsymbol{q}_1 - \cos((1-t)\frac{\phi}{2})\boldsymbol{q}_0)}{2\sin\frac{\phi}{2}}$ | $\boldsymbol{\omega} = 2\log(\boldsymbol{q}_0^{-1} \otimes \boldsymbol{q}_1)$ |
| **Euler Solver** | Update | $\boldsymbol{R}_{t+\Delta t} = \boldsymbol{R}_t \exp_M(\Delta t \cdot \boldsymbol{\Omega})$ | $\boldsymbol{q}_{t+\Delta t} = \boldsymbol{q}_t + \Delta t \cdot \boldsymbol{\eta}_t$ | $\boldsymbol{q}_{t+\Delta t} = \boldsymbol{q}_t \otimes \exp(\frac{1}{2}\Delta t \cdot \boldsymbol{\omega})$ |
| | No Renomalization | ✓ | ✗ | ✓ |
| **Numerical Stability** | $\phi \geq \pi - 10^{-2}$ | ✗ | ✓ | ✓ |
| | $\phi \leq 10^{-6}$ | ✓ | ✗ | ✓ |
| **Application Scenarios** | | FrameFlow (Yim et al., 2023a), FoldFlow (Bose et al., 2024) | AssembleFlow (Guo et al., 2025) | QFlow (**Ours**), ReQFlow (**Ours**) |

**Theorem 3.2.** *(**Reducing transport costs**). The pair $(\boldsymbol{q}_0', \boldsymbol{q}_1')$ yields lower or equal convex transport costs than the input $(\boldsymbol{q}_0, \boldsymbol{q}_1)$. For any convex $c: \mathbb{R}^3 \to \mathbb{R}$, define the cost as $C(\boldsymbol{q}_0, \boldsymbol{q}_1) = c\left(\log(\boldsymbol{q}_0^{-1} \otimes \boldsymbol{q}_1)\right)$. Then, we have $\mathbb{E}[C(\boldsymbol{q}_0', \boldsymbol{q}_1')] \leq \mathbb{E}[C(\boldsymbol{q}_0, \boldsymbol{q}_1)].$*

Theorem 3.2 shows that the coupling $(\boldsymbol{q}_0', \boldsymbol{q}_1')$ either achieves a strictly lower or the same convex transport cost compared to the original one, highlighting the advantage of the quaternion flow rectification in reducing the overall rotation displacement cost without compromising the marginal distribution constraints (Theorem 3.1). In addition, we have

**Corollary 3.3.** *(**Cost Reduction with Nonconstant Speed**). Suppose the geodesic interpolation $\boldsymbol{q}_t$ between $\boldsymbol{q}_0$ and $\boldsymbol{q}_1$ has a constant axis $\boldsymbol{u}$, but its speed is nonconstant in time, i.e, $\boldsymbol{\omega}_t = a(t)\boldsymbol{u}$. The quaternion flow rectification still reduces or preserves the transport cost.*

This corollary means that when applying the exponential step size scheduler (i.e., Eq. (15)), the rectification still reduces or preserves the transport cost.

### 3.4. Rationality Analysis

Most existing methods, like FrameFlow (Yim et al., 2023a) and FoldFlow (Bose et al., 2024), represent rotations as $3 \times 3$ matrices. Given two rotation matrices $\boldsymbol{R}_0$ and $\boldsymbol{R}_1$, they construct a flow in SO(3) with matrix geodesic interpolation:

$$\boldsymbol{R}_t = \boldsymbol{R}_0 \exp_M \left( t \log_M(\boldsymbol{R}_0^\top \boldsymbol{R}_1) \right), \qquad (16)$$

where $\exp_M(\cdot)$ and $\log_M(\cdot)$ denote the matrix exponential and logarithmic maps, respectively. The corresponding angular velocity $\boldsymbol{\Omega} = \log_M(\boldsymbol{R}_0^\top \boldsymbol{R}_1)$. Different from existing methods (Yim et al., 2023a;b; Bose et al., 2024), our method applies quaternion-based rotation representation and achieves rotation interpolation by SLERP in an exponential format, which achieves superior numerical stability and thus benefits protein backbone generation.

To verify this claim, we conduct a round-trip error experiment: given an rotation $\boldsymbol{\omega}$ in the axis-angle format, we convert it to a rotation matrix $\boldsymbol{R}$ and a quaternion $\boldsymbol{q}$, respectively, and convert it back to the axis-angle format, denoted as $\hat{\boldsymbol{\omega}}_R$ and $\hat{\boldsymbol{\omega}}_q$, respectively. Figure 2a shows the round-trip errors in $L_2$ norm for large rotation angles (e.g., $\phi \in [\pi - 10^{-2}, \pi)$). Our quaternion-based method is numerically stable while the matrix-based representation suffers

severe numerical errors. When training a protein back-bone generation model, the numerical stability for large rotation angles is important. Given the frames in the Protein Data Bank (PDB) (Burley et al., 2023) dataset and the SCOPe (Chandonia et al., 2022) dataset, we sample a random noise for each frame and calculate the rotation angle between them. The histogram in Figure 2b shows that when training an arbitrary flow-based model, the probability of suffering at least one large angle per protein is 0.59 for PDB and 0.34 for SCOPe, respectively. It means that the matrix-based representation may introduce undesired numerical errors that aggregate and propagate during training.

In addition, a very recent work, AssembleFlow (Guo et al., 2025), also applies quaternion-based rotation representation and SLERP when modeling 3D molecules. In particular, it applies SLERP in an additive format:

$$q_t = \frac{\sin((1-t)\frac{\phi}{2})}{\sin(\frac{\phi}{2})} q_0 + \frac{\sin(t\frac{\phi}{2})}{\sin(\frac{\phi}{2})} q_1, \qquad (17)$$

and updates rotations linearly by the following Euler solver:

$$q_{t+\Delta t} = q_t + \Delta t \cdot \eta_t. \qquad (18)$$

Here, $\eta_t$ is the instantaneous velocity in the tangent space of $q_t$, which is derived by the first-order derivative of Eq. (17). However, this modeling strategy also suffers numerical issues. Firstly, although the additive format SLERP can generate the same interpolation path as ours in theory, when rotation angle $\phi$ is small (e.g., $\phi \in [0, 10^{-6})$), Eq. (17) often outputs "NaN" because the denominator $\sin(\frac{\phi}{2})$ tends to zero. The exponential step size scheduler leads to rapid convergence when generating protein backbones, which frequently generates rotation angles below the threshold $10^{-6}$ (as shown in Figure 2c) and thus makes the additive format SLERP questionable in our task. Secondly, the Euler step in Eq. (18) makes $\|q_{t+\Delta t}\|_2 \neq 1$, so that renormalization is required after each update. Table 1 provides a comprehensive comparison for the three rotation interpolation methods, highlighting the advantages of our method.

# 4. Experiment

To demonstrate the effectiveness and efficiency of our methods (QFlow and ReQFlow), we conduct comprehensive experiments to compare them with state-of-the-art protein backbone generation methods. In addition, we conduct ablation studies to verify the usefulness of the flow rectification strategy and the impact of sampling steps on model performance. All the experiments are implemented on four NVIDIA A100 80G GPUs. Implementation details and experimental results are shown in this section and Appendix C.

*Table 2.* Comparisons for various models on PDB. For each designability metric, we bold the best result and show the top-3 results with a blue background. In the same way, we indicate the best and top-3 diversity and novelty results among the rows with Fraction $> 0.8$. The inference time corresponds to generating a protein backbone with length $N = 300$.

| Method | Efficiency | | Designability | | Diversity | Novelty |
|---|---|---|---|---|---|---|
| | Step | Time(s) | Fraction↑ | scRMSD↓ | TM↓ | TM↓ |
| RFDiffusion | 50 | 66.23 | 0.904 | $1.102_{\pm 1.617}$ | 0.382 | 0.527 |
| Genie2 | 1000 | 112.93 | 0.908 | $1.132_{\pm 1.389}$ | 0.370 | **0.475** |
| | 500 | 55.86 | 0.000 | $18.169_{\pm 5.963}$ | - | - |
| FrameDiff | 500 | 48.12 | 0.564 | $2.936_{\pm 3.093}$ | 0.441 | 0.591 |
| FoldFlow$_{Base}$ | 500 | 43.52 | 0.624 | $3.080_{\pm 3.449}$ | 0.469 | 0.645 |
| FoldFlow$_{SFM}$ | 500 | 43.63 | 0.636 | $3.031_{\pm 3.589}$ | 0.411 | 0.604 |
| FoldFlow$_{OT}$ | 500 | 43.35 | 0.852 | $1.760_{\pm 2.593}$ | 0.434 | 0.617 |
| FoldFlow2 | 50 | 6.35 | 0.952 | $1.083_{\pm 1.308}$ | 0.373 | 0.527 |
| | 20 | 2.63 | 0.644 | $3.060_{\pm 3.210}$ | 0.339 | 0.492 |
| FrameFlow | 500 | 17.05 | 0.872 | $1.380_{\pm 1.392}$ | 0.346 | 0.562 |
| | 200 | 6.77 | 0.864 | $1.542_{\pm 1.889}$ | 0.348 | 0.564 |
| | 100 | 3.46 | 0.708 | $2.167_{\pm 2.373}$ | 0.332 | 0.560 |
| | 50 | 1.73 | 0.704 | $2.639_{\pm 3.079}$ | 0.334 | 0.536 |
| | 20 | 0.71 | 0.436 | $4.652_{\pm 4.390}$ | 0.319 | 0.501 |
| | 10 | 0.37 | 0.180 | $7.343_{\pm 5.125}$ | 0.317 | 0.482 |
| QFlow | 500 | 17.37 | 0.936 | $1.163_{\pm 0.938}$ | 0.356 | 0.635 |
| | 200 | 7.10 | 0.864 | $1.400_{\pm 1.259}$ | **0.344** | 0.620 |
| | 100 | 3.48 | 0.916 | $1.342_{\pm 1.364}$ | 0.348 | 0.614 |
| | 50 | 1.77 | 0.812 | $1.785_{\pm 2.151}$ | **0.344** | 0.571 |
| | 20 | 0.73 | 0.604 | $3.090_{\pm 3.374}$ | 0.325 | 0.537 |
| | 10 | 0.38 | 0.332 | $5.032_{\pm 4.303}$ | 0.313 | 0.528 |
| ReQFlow | 500 | 17.42 | **0.972** | **$1.071_{\pm 0.482}$** | 0.377 | 0.645 |
| | 200 | 6.94 | 0.932 | $1.160_{\pm 0.782}$ | 0.384 | 0.648 |
| | 100 | 3.58 | 0.928 | $1.245_{\pm 1.059}$ | 0.369 | 0.629 |
| | 50 | 1.78 | 0.912 | $1.254_{\pm 0.915}$ | 0.369 | 0.608 |
| | 20 | 0.72 | 0.872 | $1.418_{\pm 0.998}$ | 0.355 | 0.581 |
| | 10 | 0.38 | 0.676 | $2.443_{\pm 2.382}$ | 0.337 | 0.540 |

## 4.1. Experimental Setup

**Datasets.** We apply two commonly used datasets in our experiments. The first is the 23,366 protein backbones collected from Protein Data Bank (PDB) (Burley et al., 2023), whose lengths range from 60 to 512. The second is the SCOPe dataset (Chandonia et al., 2022) pre-processed by FrameFlow (Yim et al., 2023a), which contains 3,673 protein backbones with lengths ranging from 60 to 128.

**Baselines.** The baselines of our methods include diffusion-based methods (FrameDiff (Yim et al., 2023b), RFDiffusion (Watson et al., 2023), and Genie2 (Lin et al., 2024)) and flow-based methods (FrameFlow (Yim et al., 2023a), FoldFlow (Bose et al., 2024), and FoldFlow2 (Huguet et al., 2024)). In addition, we rectify FrameFlow by our method (i.e., re-training FrameFlow based on the paired data generated by itself) and consider the rectified FrameFlow (Re-FrameFlow) as a baseline as well.

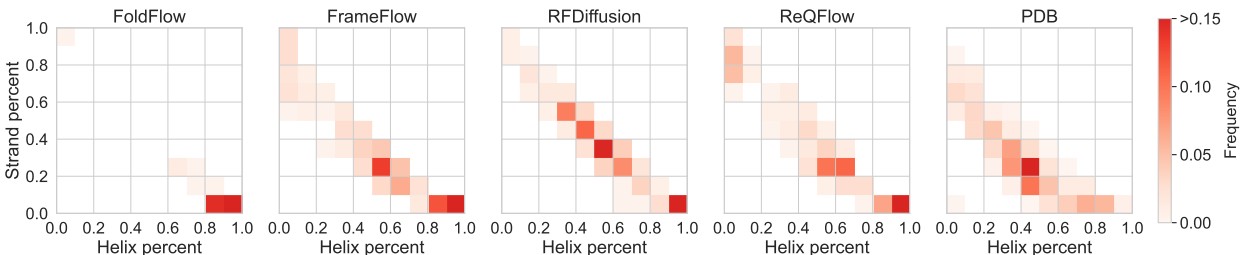

*Figure 3.* The distribution of protein backbones with respect to the percentages of their secondary structure.

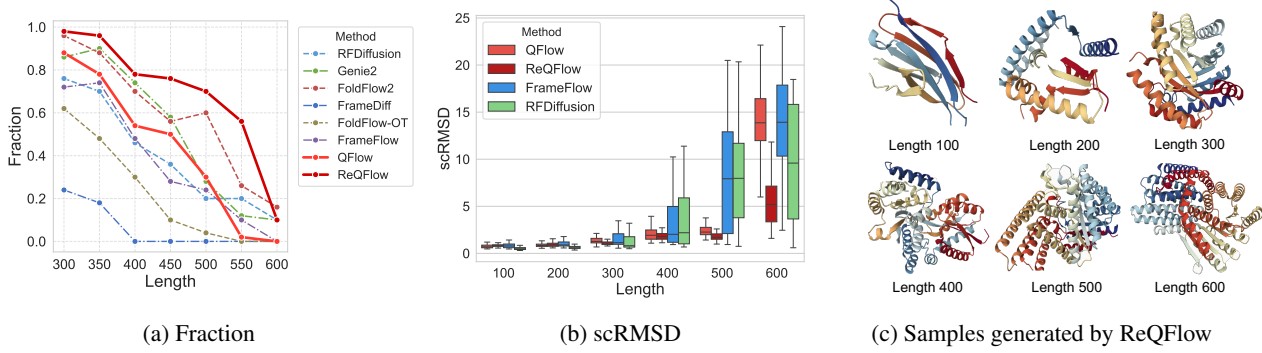

| (a) Fraction | (b) scRMSD | (c) Samples generated by ReQFlow |

*Figure 4.* The comparison for various methods on the designability of generated long-chain protein backbones.

**Implementation Details.** For the PDB dataset, we utilize the checkpoints of baselines and reproduce the results shown in their papers. Given the QFlow trained on PDB, we generate 7,653 protein backbones with lengths in $[60, 512]$ from noise and then train ReQFlow based on these noise-backbone pairs. For the SCOPe dataset, we train all the models from scratch. Given the QFlow trained on SCOPe, we generate 3,167 protein backbones with lengths in $[60, 128]$ from noise and then train ReQFlow based on these noise-backbone pairs. When training ReQFlow, we apply structural data filtering, selecting training samples based on scRMSD ($\leq 2$Å) and TM-score ($\geq 0.9$ for long-chain proteins) and removing proteins with excessive loops (>50%) or abnormally large radius of gyration (top 4%). ReFrameFlow is trained in the same way.

**Evaluation Metrics.** Following previous works (Yim et al., 2023a; Bose et al., 2024; Geffner et al., 2025), we evaluate each method in the following four aspects:

**1) Designability:** As the most critical metric, designability reflects the possibility that a generated protein backbone can be realized by folding the corresponding amino acid sequence. It is assessed by the scRMSD between the generated protein backbone and the backbone predicted by ESMFold (Lin et al., 2023). Given a set of generated backbones, we calculate the proportion of the backbones whose scRMSD $\leq 2$Å (denoted as Fraction).

**2) Diversity:** Given designable protein backbones, whose scRMSD $\leq 2$Å, we quantify structural diversity by averaging the mean pairwise TM-scores computed for each backbone length.

**3) Novelty:** For each designable protein backbone, we compute its maximum TM-score to the data in PDB using Foldseek (van Kempen et al., 2022). The average of the scores reflect the novelty of the generated protein backbones.

**4) Efficiency:** We assess the computational efficiency of each method by the number of sampling steps and the average inference time for generating 50 proteins at two lengths: 300 residues for PDB and 128 residues for SCOPe.

### 4.2. Comparison Experiments on PDB

**Generation Quality.** Given the models trained on PDB, we set the length of backbone $N \in \{100, 150, 200, 250, 300\}$, and generate 50 protein backbones for each length. Table 2 shows that ReQFlow achieves state-of-the-art performance in designability, achieving the highest Fraction (0.972) among all models, significantly outperforming strong competitors such as Genie2 (0.908) and RFDiffusion (0.904). Additionally, it achieves the lowest scRMSD (1.071±0.482), with a notably smaller variance compared to the other methods, highlighting the model's consistency and reliability in generating high-quality protein backbones. Meanwhile, Re-

QFlow maintains competitive performance in diversity and novelty.

**Computational Efficiency.** Moreover, ReQFlow achieves ultra-fast protein backbone generation. Typically, ReQFlow achieves a high Fraction score (0.912) with merely 50 steps and 1.78s, outperforming RFDiffusion and Genie2 with $37\times$ and $63\times$ acceleration, respectively. The state-of-the-art methods like Genie2 and FoldFlow2 suffer severe performance degradation in designability when the number of steps is halved, while ReQFlow performs stably even reducing the number of steps from 500 to 20. In addition, although the main speed bottleneck is model prediction, ReQFlow's rotation update can be $\sim$20% faster than Frame-Flow's update because of utilizing the quaternion-based computation (Appendix C.3).

**Fitness of Data Distribution.** Given generated protein backbones, we record the percentages of helix and strand, respectively, for each backbone, and visualize the distribution of the backbones with respect to the percentages in Figure 3. The protein backbones generated by ReQFlow have a reasonable distribution, which is similar to those of RFDiffusion and FrameFlow and comparable to that of the PDB dataset. However, the distribution of FoldFlow is significantly different from the data distribution and indicates a mode collapse risk — the protein backbones generated by FoldFlow are always dominated by helix structures.

**Effectiveness on Long Chain Generation.** Notably, Re-QFlow demonstrates exceptional performance in generating long-chain protein backbones (e.g., $N > 300$). As shown in Figures 4a and 4b, ReQFlow outperforms all baselines on generating long protein backbones and shows remarkable robustness. Especially, when the length $N > 500$, which is out of the length range of PDB data, all the baselines fail to maintain high designability while ReQFlow still achieves promising performance in Fraction score and scRMSD and generates reasonable protein backbones. This generalization ability beyond the training data distribution underscores ReQFlow's potential for real-world applications requiring robust long-chain protein design.

**Ablation Study.** We conduct an ablation study to evaluate the impact of different components in the ReQFlow model. The results in Table 3 reveal that similar to existing methods (Yim et al., 2023a; Bose et al., 2024; Huguet et al., 2024), the exponential step size scheduler is important for ReQFlow, helping generate designable protein backbones with relatively few steps. Additionally, the data filter is necessary for making flow rectification work. In particular, rectifying QFlow based on low-quality data leads to a substantial degradation in model performance. In contrast, after filtering out noisy and irrelevant data, rectifying QFlow based on the high-quality data boosts the model performance significantly.

*Table 3.* The Fraction scores of ReQFlow under different settings when generating backbones with 300 residues by 500 and 50 steps. For each method, we evaluate the results on five checkpoints.

| Exponential Scheduler | Flow Rectification | Data Filtering | Sampling Steps 500 | 50 |
|---|---|---|---|---|
| ✗ | ✗ | ✗ | $0.143_{\pm 0.079}$ | $0.047_{\pm 0.030}$ |
| ✓ | ✗ | ✗ | $0.910_{\pm 0.029}$ | $0.795_{\pm 0.051}$ |
| ✓ | ✓ | ✗ | $0.612_{\pm 0.084}$ | $0.519_{\pm 0.154}$ |
| ✓ | ✓ | ✓ | $0.969_{\pm 0.027}$ | $0.932_{\pm 0.022}$ |

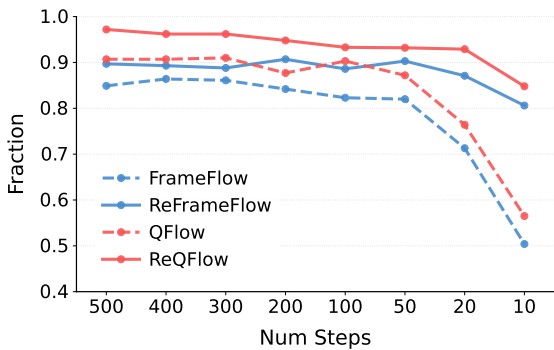

*Figure 5.* A comparison for various methods on their designability with the reduction of sampling steps. Original data is in Table 8.

### 4.3. Analytic Experiments on SCOPe

**Universality of Flow Rectification.** Note that, the flow rectification method used in our work is universal for various models. As shown in Table 4 and Figure 5, applying flow rectification, we can improve the efficiency and effectiveness of FrameFlow as well. This result highlights the broad utility of flow rectification as an operation that can enhance the performance of flow models on SO(3) spaces.

**Superiority of Exponential-Format SLERP.** The results in Table 4 and Figure 5 indicate that QFlow and ReQFlow outperform their corresponding counterparts (FrameFlow and ReFrameFlow) in terms of designability across all sampling steps. As we analyzed in Section 3.4, the superiority of our models can be attributed to the better numerical stability of quaternion calculations compared to the traditional matrix geodesic method.

## 5. Conclusion and Future Work

In this study, we propose a rectified quaternion flow matching method for efficient and high-quality protein backbone generation. Leveraging quaternion-based representation and flow rectification, our method achieves encouraging performance and significantly reduces inference time. In the near future, we plan to improve our method for generating high-quality long-chain protein backbones. This

*Table 4.* Comparisons for various models on SCOPe. Five checkpoints per model are evaluated to show statistical significance. We indicate the best and top-3 results in the same way as Table 2 does.

| | Step | Epoch | Designability | | Diversity | Novelty | | Step | Epoch | Designability | | Diversity | Novelty |
|---|---|---|---|---|---|---|---|---|---|---|---|---|---|
| | | | Fraction↑ | scRMSD↓ | TM↓ | TM↓ | | | | Fraction↑ | scRMSD↓ | TM↓ | TM↓ |
| **FrameFlow** | 500 | 187 | 0.880 | $1.418_{\pm1.155}$ | 0.399 | 0.711 | **QFlow** | 500 | 181 | 0.899 | $1.240_{\pm0.781}$ | 0.392 | 0.695 |
| | | 189 | 0.849 | $1.448_{\pm1.114}$ | 0.397 | 0.713 | | | 185 | 0.913 | $1.210_{\pm0.792}$ | 0.393 | 0.719 |
| | | 193 | 0.851 | $1.396_{\pm1.167}$ | 0.390 | 0.720 | | | 195 | 0.907 | $1.263_{\pm1.334}$ | 0.389 | 0.712 |
| | | 195 | 0.845 | $1.427_{\pm1.011}$ | 0.397 | 0.713 | | | 197 | 0.893 | $1.285_{\pm0.929}$ | 0.401 | 0.716 |
| | | 199 | 0.830 | $1.498_{\pm1.075}$ | 0.379 | 0.712 | | | 199 | 0.852 | $1.444_{\pm0.991}$ | 0.385 | 0.693 |
| | 50 | 187 | 0.797 | $1.603_{\pm1.138}$ | 0.382 | 0.686 | | 50 | 181 | 0.849 | $1.447_{\pm1.148}$ | 0.379 | **0.648** |
| | | 189 | 0.820 | $1.546_{\pm1.316}$ | 0.379 | 0.685 | | | 185 | 0.875 | $1.389_{\pm1.169}$ | 0.386 | 0.687 |
| | | 193 | 0.836 | $1.504_{\pm1.158}$ | 0.375 | 0.680 | | | 195 | 0.872 | $1.389_{\pm1.314}$ | **0.371** | 0.674 |
| | | 195 | 0.817 | $1.528_{\pm1.019}$ | 0.390 | 0.682 | | | 197 | 0.823 | $1.517_{\pm1.196}$ | 0.384 | 0.682 |
| | | 199 | 0.787 | $1.650_{\pm1.295}$ | 0.366 | 0.676 | | | 199 | 0.849 | $1.461_{\pm0.970}$ | 0.378 | 0.682 |
| | 20 | 187 | 0.739 | $1.888_{\pm1.452}$ | 0.373 | 0.656 | | 20 | 181 | 0.778 | $1.746_{\pm1.462}$ | 0.369 | 0.620 |
| | | 189 | 0.713 | $1.918_{\pm1.495}$ | 0.362 | 0.656 | | | 185 | 0.778 | $1.683_{\pm1.359}$ | 0.377 | 0.655 |
| | | 193 | 0.723 | $1.882_{\pm1.488}$ | 0.371 | 0.645 | | | 195 | 0.764 | $1.764_{\pm1.529}$ | 0.367 | 0.646 |
| | | 195 | 0.687 | $2.035_{\pm1.574}$ | 0.377 | 0.650 | | | 197 | 0.746 | $1.834_{\pm1.511}$ | 0.374 | 0.652 |
| | | 199 | 0.677 | $2.106_{\pm1.770}$ | 0.369 | 0.636 | | | 199 | 0.742 | $1.802_{\pm1.256}$ | 0.373 | 0.648 |
| **ReFrameFlow** | 500 | 2 | 0.912 | $1.205_{\pm0.675}$ | 0.406 | 0.713 | **ReQFlow** | 500 | 2 | 0.939 | $1.120_{\pm0.647}$ | 0.411 | 0.704 |
| | | 3 | 0.933 | $1.211_{\pm0.631}$ | 0.410 | 0.713 | | | 3 | 0.952 | $\mathbf{1.088}_{\pm0.523}$ | 0.404 | 0.703 |
| | | 4 | 0.923 | $1.201_{\pm0.600}$ | 0.411 | 0.709 | | | 4 | 0.939 | $1.149_{\pm0.595}$ | 0.405 | 0.693 |
| | | 5 | 0.923 | $1.270_{\pm0.721}$ | 0.399 | 0.701 | | | 5 | 0.949 | $1.155_{\pm0.565}$ | 0.402 | 0.690 |
| | | 6 | 0.930 | $1.178_{\pm0.628}$ | 0.407 | 0.709 | | | 6 | **0.955** | $1.143_{\pm0.819}$ | 0.407 | 0.688 |
| | 50 | 2 | 0.903 | $1.267_{\pm0.704}$ | 0.407 | 0.691 | | 50 | 2 | 0.928 | $1.157_{\pm0.747}$ | 0.411 | 0.687 |
| | | 3 | 0.897 | $1.262_{\pm0.739}$ | 0.408 | 0.692 | | | 3 | 0.919 | $1.179_{\pm0.762}$ | 0.410 | 0.688 |
| | | 4 | 0.909 | $1.271_{\pm0.674}$ | 0.408 | 0.690 | | | 4 | 0.916 | $1.194_{\pm0.850}$ | 0.413 | 0.676 |
| | | 5 | 0.906 | $1.270_{\pm0.717}$ | 0.406 | 0.690 | | | 5 | 0.933 | $1.184_{\pm0.691}$ | 0.407 | 0.670 |
| | | 6 | 0.914 | $1.272_{\pm0.868}$ | 0.406 | 0.684 | | | 6 | 0.914 | $1.229_{\pm0.768}$ | 0.414 | 0.681 |
| | 20 | 2 | 0.877 | $1.432_{\pm1.043}$ | 0.400 | 0.669 | | 20 | 2 | 0.929 | $1.267_{\pm0.844}$ | 0.407 | 0.678 |
| | | 3 | 0.878 | $1.378_{\pm0.794}$ | 0.410 | 0.673 | | | 3 | 0.913 | $1.256_{\pm0.760}$ | 0.404 | 0.674 |
| | | 4 | 0.899 | $1.354_{\pm0.810}$ | 0.405 | 0.679 | | | 4 | 0.913 | $1.232_{\pm0.725}$ | 0.406 | 0.671 |
| | | 5 | 0.877 | $1.440_{\pm0.936}$ | 0.400 | 0.672 | | | 5 | 0.893 | $1.322_{\pm0.719}$ | 0.404 | 0.670 |
| | | 6 | 0.888 | $1.393_{\pm1.023}$ | 0.409 | 0.675 | | | 6 | 0.900 | $1.331_{\pm0.885}$ | 0.405 | 0.663 |

will involve constructing a larger training dataset, building on approaches such as Genie2 (Lin et al., 2024) and Proteína (Geffner et al., 2025). Additionally, we plan to refine our model architecture through two key strategies: increasing model capacity via parameter scaling and exploring non-equivariant design, drawing inspiration from the architecture of Proteína (Geffner et al., 2025). Furthermore, we intend to leverage the knowledge embedded in large-scale pre-training models (Li et al., 2025; Huguet et al., 2024), such as FoldFlow2 (Huguet et al., 2024), which incorporated sequence information using ESM2 (Lin et al., 2023). As long-term goals, we will extend our method to conditional protein backbone generation and explore its applications in side-chain generation and full-atom protein generation.

## Acknowledgements

This work was supported by the National Natural Science Foundation of China (92270110), the Fundamental Research Funds for the Central Universities, the Research Funds of Renmin University of China, and the Public Computing Cloud, Renmin University of China. We also acknowledge the support provided by the fund for building world-class universities (disciplines) of Renmin University of China and by the funds from Beijing Key Laboratory of Research on Large Models and Intelligent Governance, Engineering Research Center of Next-Generation Intelligent Search and Recommendation, Ministry of Education, and from Intelligent Social Governance Interdisciplinary Platform, Major Innovation & Planning Interdisciplinary Platform for the "Double-First Class" Initiative, Renmin University of China.

## Impact Statement

This paper presents work whose aim is to advance the field of Machine Learning and AI for Science, especially the task of protein design. There are many potential societal consequences of our work, e.g., accelerating drug development and contributing to healthcare. None of them we feel must be specifically highlighted here.

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

# A. Proofs of Key Theoretical Results

## A.1. The Angular Velocity under Exponential Scheduler

**Proposition A.1.** *For spherical linear interpolation (SLERP) with angular velocity $\boldsymbol{\omega}$, when applying an exponential scheduler during inference:*

$$\boldsymbol{q}_t = \boldsymbol{q}_0 \otimes \exp\left((1 - e^{-\gamma t}) \log(\boldsymbol{q}_0^{-1} \otimes \boldsymbol{q}_1)\right), \tag{19}$$

*the resulting angular velocity evolves as $\hat{\boldsymbol{\omega}}_t = \gamma e^{-\gamma t}\boldsymbol{\omega}$.*

*Proof.* The standard SLERP formulation in exponential form is:

$$\boldsymbol{q}_t = \boldsymbol{q}_0 \otimes \exp\left(t \log(\boldsymbol{q}_0^{-1} \otimes \boldsymbol{q}_1)\right), \tag{20}$$

where the relative rotation $\boldsymbol{q}_{\text{rel}} = \boldsymbol{q}_0^{-1} \otimes \boldsymbol{q}_1$ has logarithm map $\log(\boldsymbol{q}_{\text{rel}}) = \frac{1}{2}\phi\boldsymbol{u}$. The angular velocity is:

$$\boldsymbol{\omega} = 2 \cdot \log(\boldsymbol{q}_{\text{rel}}) = \phi\boldsymbol{u}. \tag{21}$$

Introducing an exponential scheduler $\kappa(t) = 1 - e^{-\gamma t}$ with derivative $\kappa'(t) = \gamma e^{-\gamma t}$, the modified SLERP becomes:

$$\boldsymbol{q}_t = \boldsymbol{q}_0 \otimes \exp\left(\kappa(t) \log(\boldsymbol{q}_{\text{rel}})\right). \tag{22}$$

Differentiating with respect to time using the chain rule:

$$\begin{aligned}
\dot{\boldsymbol{q}}_t &= \boldsymbol{q}_0 \otimes \frac{d}{dt} \exp\left(\kappa(t) \log(\boldsymbol{q}_{\text{rel}})\right) \\
&= \gamma e^{-\gamma t} \log(\boldsymbol{q}_{\text{rel}}) \otimes \boldsymbol{q}_0 \otimes \exp\left(\kappa(t) \log(\boldsymbol{q}_{\text{rel}})\right) \\
&= \gamma e^{-\gamma t} \log(\boldsymbol{q}_{\text{rel}}) \otimes \boldsymbol{q}_t.
\end{aligned} \tag{23}$$

Applying the quaternion kinematics equation $\dot{\boldsymbol{q}} = \frac{1}{2}[0, \boldsymbol{\omega}^\top]^\top \otimes \boldsymbol{q}$ (Sola, 2017), we solve for the effective angular velocity:

$$\begin{aligned}
[0, \hat{\boldsymbol{\omega}}_t^\top]^\top &= 2\dot{\boldsymbol{q}}_t \otimes \boldsymbol{q}_t^{-1} \\
&= 2\gamma e^{-\gamma t} \log(\boldsymbol{q}_{\text{rel}}) \otimes \boldsymbol{q}_t \otimes \boldsymbol{q}_t^{-1} \\
&= 2\gamma e^{-\gamma t} \log(\boldsymbol{q}_{\text{rel}}).
\end{aligned} \tag{24}$$

Substituting the angular velocity from Eq. (21) yields:

$$\hat{\boldsymbol{\omega}}_t = \gamma e^{-\gamma t}\boldsymbol{\omega}. \tag{25}$$

$\square$

## A.2. Proofs of The Theorems in Section 3.3

Our proofs yield the same pipeline used in (Liu, 2022). The proofs are inspired by that work and derived based on the same techniques. What we did is extending and specifying the theoretical results in (Liu, 2022) for $\mathbb{S}^3$. The original rotation process is $\{\boldsymbol{q}_t\}_{t \in [0,1]}$, where each $\boldsymbol{q}_t$ is a unit quaternion representing a rotation in SO(3), $\boldsymbol{\omega}_t \in \mathbb{R}^3$ is the angular velocity at time $t$. The quaternion dynamics are given by

$$\dot{\boldsymbol{q}}_t = \frac{1}{2}[0, \boldsymbol{\omega}_t^\top]^\top \otimes \boldsymbol{q}_t \in T_{\boldsymbol{q}_t}(\mathbb{S}^3), \tag{26}$$

where $T_{\boldsymbol{q}_t}(\mathbb{S}^3)$ is the tangent space at $\boldsymbol{q}_t$. We write $\boldsymbol{q}_0 \sim \mathcal{Q}_0$, $\boldsymbol{q}_1 \sim \mathcal{Q}_1$ for the initial and target distributions. For a given input coupling $(\boldsymbol{q}_0, \boldsymbol{q}_1)$, the exact minimum of $\mathcal{L}_{\text{SO}(3)}$ in Eq. (11) is achieved if

$$\tilde{\boldsymbol{\omega}}_{\theta,t} = \tilde{\boldsymbol{\omega}}_t(\boldsymbol{q}, t) = \mathbb{E}[\boldsymbol{\omega}_t | \boldsymbol{q}_t = \boldsymbol{q}] \in \mathbb{R}^3, \tag{27}$$

which is the expected angular velocity at point $\boldsymbol{q}$, time $t$. We now define the rectified process $\{\boldsymbol{q}'_t\}_{t \in [0,1]}$ by

$$\dot{\boldsymbol{q}}'_t = \frac{1}{2}[0, \tilde{\boldsymbol{\omega}}_t(\boldsymbol{q}'_t, t)^\top]^\top \otimes \boldsymbol{q}'_t, \quad \boldsymbol{q}'_0 \sim \mathcal{Q}_0, \tag{28}$$

### A.2.1. PROOF OF THEOREMS 3.1

*Proof.* Consider any smooth test function $h : \mathbb{S}^3 \to \mathbb{R}$. By chain rule:

$$\frac{d}{dt}\mathbb{E}[h(\mathbf{q}_t)] = \mathbb{E}\big[\nabla_{\mathbb{S}^3} h(\boldsymbol{q}_t) \cdot \dot{\boldsymbol{q}}_t\big], \tag{29}$$

where $\nabla_{\mathbb{S}^3} h$ is the gradient on the manifold. From the definition in Eq. (26), since $\boldsymbol{\omega}_t$ is random, we rewrite inside the expectation by conditioning on $\boldsymbol{q}_t$:

$$\mathbb{E}\big[\nabla_{\mathbb{S}^3} h(\boldsymbol{q}_t) \cdot \dot{\boldsymbol{q}}_t\big] = \mathbb{E}\Big[\nabla_{\mathbb{S}^3} h(\boldsymbol{q}_t) \cdot \tfrac{1}{2}\big[0, \mathbb{E}(\boldsymbol{\omega}_t|\boldsymbol{q}_t)^\top\big]^\top \otimes \boldsymbol{q}_t\Big], \tag{30}$$

because $\boldsymbol{\omega}_t|(\boldsymbol{q}_t = \boldsymbol{q})$ has conditional mean $\tilde{\boldsymbol{\omega}}_t(\boldsymbol{q}, t)$,

$$\frac{d}{dt}\mathbb{E}[h(\boldsymbol{q}_t)] = \mathbb{E}\big[\nabla_{\mathbb{S}^3} h(\boldsymbol{q}_t) \cdot \tfrac{1}{2}[0, \tilde{\boldsymbol{\omega}}_t(\boldsymbol{q}_t, t)^\top]^\top \otimes \boldsymbol{q}_t\big]. \tag{31}$$

This evolution is exactly the *weak (distributional) form* of the continuity equation:

$$\partial_t \, \mu_t + \nabla \cdot \big(\tfrac{1}{2}[0, \tilde{\boldsymbol{\omega}}_t(\boldsymbol{q}, t)^\top]^\top \otimes \boldsymbol{q} \cdot \mu_t\big) = 0, \tag{32}$$

where $\mu_t = \text{Law}(\boldsymbol{q}_t)$. According to Eq. (28), That is exactly the same weak-form evolution equation satisfied by the $\boldsymbol{q}_t'$ process, where $\boldsymbol{\omega}$ is simply replaced by $\tilde{\boldsymbol{\omega}}_t$. If we let $\nu_t := \text{Law}(\boldsymbol{q}_t')$, it solves the same continuity equation with the same initial data $\nu_0 = \mu_0$. On a compact manifold like $SO(3)$, the continuity equation has a unique solution given an initial distribution. Hence $\mu_t = \nu_t$ at all times $t$. That is,

$$\text{Law}(\boldsymbol{q}_t') = \text{Law}(\boldsymbol{q}_t), \quad \text{for all } t \in [0, 1]. \tag{33}$$

$\square$

### A.2.2. PROOF OF THEOREMS 3.2

*Proof.* The net rotation from $\boldsymbol{q}_0$ to $\boldsymbol{q}_1$ can be given by integrating the angular velocity $\boldsymbol{\omega}_t \in \mathbb{R}^3$.

$$\log\big(\boldsymbol{q}_0^{-1} \otimes \boldsymbol{q}_1\big) = \frac{1}{2} \int_0^1 \boldsymbol{\omega}_t \, dt, \tag{34}$$

and similarly,

$$\log\big(\boldsymbol{q}_0'^{-1} \otimes \boldsymbol{q}_1'\big) = \frac{1}{2} \int_0^1 \tilde{\boldsymbol{\omega}}_t(\boldsymbol{q}_t', t) \, dt, \tag{35}$$

Strictly speaking, one must keep track of the axis direction to ensure consistency, but the geodesic assumption here handles that. The rectified angular velocity $\tilde{\boldsymbol{\omega}}_t = \mathbb{E}[\boldsymbol{\omega}_t|\boldsymbol{q}_t]$ implies that the total rotation in the rectified process is a conditional expectation of the original rotation:

$$\log\big(\boldsymbol{q}_0'^{-1} \otimes \boldsymbol{q}_1'\big) = \frac{1}{2} \int_0^1 \tilde{\boldsymbol{\omega}}_t \, dt = \frac{1}{2}\mathbb{E}\left[\int_0^1 \boldsymbol{\omega}_t \, dt \,\middle|\, \{\boldsymbol{q}_t'\}\right]. \tag{36}$$

Applying Jensen's inequality to the convex cost $c$ over this conditional expectation:

$$c\big(\log\big(\boldsymbol{q}_0'^{-1} \otimes \boldsymbol{q}_1'\big)\big) = c\left(\frac{1}{2}\mathbb{E}\left[\int_0^1 \boldsymbol{\omega}_t \, dt \,\middle|\, \{\boldsymbol{q}_t'\}\right]\right) \leq \mathbb{E}\left[\frac{1}{2}c\left(\int_0^1 \boldsymbol{\omega}_t \, dt\right) \,\middle|\, \{\boldsymbol{q}_t'\}\right]. \tag{37}$$

Taking the total expectation on both sides:

$$\mathbb{E}\left[C(\boldsymbol{q}_0', \boldsymbol{q}_1')\right] \leq \mathbb{E}\left[\frac{1}{2}c\left(\int_0^1 \boldsymbol{\omega}_t \, dt\right)\right] = \mathbb{E}\left[C(\boldsymbol{q}_0, \boldsymbol{q}_1)\right]. \tag{38}$$

This final inequality establishes that the rectified coupling $(\boldsymbol{q}_0', \boldsymbol{q}_1')$ achieves equal or lower expected transport cost than the original coupling $(\boldsymbol{q}_0, \boldsymbol{q}_1)$.

$\square$

### A.2.3. PROOF OF COROLLARY 3.3

*Proof.* Suppose the original process has the nonconstant angular velocity $\boldsymbol{\omega}_t = a(t)\boldsymbol{u}$ (fixed axis), with $\tau = \frac{1}{2}\int_0^1 a(t)dt$.

$$\log\big(\boldsymbol{q}_0^{-1} \otimes \boldsymbol{q}_1\big) = \frac{1}{2}\int_0^1 \boldsymbol{\omega}_t \, dt = \frac{1}{2}\boldsymbol{u}\int_0^1 a(t)\, dt = \tau\boldsymbol{u} \tag{39}$$

Recall that the rectified angular velocity is:

$$\tilde{\boldsymbol{\omega}}_t(\boldsymbol{q}, t) = \mathbb{E}[\boldsymbol{\omega}_t|\boldsymbol{q}_t] \tag{40}$$

Since $\boldsymbol{\omega}_t = a(t)\boldsymbol{u}$, we simply get:

$$\tilde{\boldsymbol{\omega}}_t(\boldsymbol{q}, t) = \mathbb{E}[a(t)|\boldsymbol{q}_t]\boldsymbol{u} \tag{41}$$

The total rotation from $\boldsymbol{q}_0'$ to the $\boldsymbol{q}_1'$ in the rectified process satisfies:

$$\log(\boldsymbol{q}_0'^{-1} \otimes \boldsymbol{q}_1') = \frac{1}{2}\int_0^1 \tilde{\boldsymbol{\omega}}_t(\boldsymbol{q}_t', t)dt = \frac{1}{2}\left(\int_0^1 \mathbb{E}[a(t) \,|\, \boldsymbol{q}_t']dt\right)\boldsymbol{u}. \tag{42}$$

Let $\tau' = \frac{1}{2}\int_0^1 \mathbb{E}[a(t) \,|\, \boldsymbol{q}_t']dt$. Thus,

$$\log(\boldsymbol{q}_0'^{-1} \otimes \boldsymbol{q}_1') = \tau'\boldsymbol{u} \tag{43}$$

Because $\tau = \frac{1}{2}\int_0^1 a(t)dt$, $\tau' = \frac{1}{2}\int_0^1 \mathbb{E}[a(t) \,|\, \boldsymbol{q}_t]dt$, and Eq. (36) in Theorem 3.2, we note

$$\tau'\boldsymbol{u} = \frac{1}{2}\boldsymbol{u}\left(\int_0^1 \mathbb{E}[a(t) \,|\, \boldsymbol{q}_t']dt\right) = \frac{1}{2}\mathbb{E}\left[\int_0^1 a(t)\boldsymbol{u}\, dt \,\bigg|\, \{\boldsymbol{q}_t'\}\right] = \mathbb{E}[\tau\boldsymbol{u}|\{\boldsymbol{q}_t'\}] \tag{44}$$

For the coupling $(\boldsymbol{q}_0', \boldsymbol{q}_1')$, the cost is:

$$C(\boldsymbol{q}_0', \boldsymbol{q}_1') = c(\tau'\boldsymbol{u}). \tag{45}$$

Since $\tau'\boldsymbol{u} = \mathbb{E}[\tau\boldsymbol{u}|\{\boldsymbol{q}_t'\}]$, convexity of $c$ implies Jensen's inequality in conditional form:

$$c(\tau'\boldsymbol{u}) = c(\mathbb{E}[\tau\boldsymbol{u}|\{\boldsymbol{q}_t'\}]) \le \mathbb{E}[c(\tau\boldsymbol{u})|\{\boldsymbol{q}_t'\}] \tag{46}$$

Next, take unconditional expectation on both sides. By the law of total expectation (tower property),

$$\mathbb{E}[c(\tau'\boldsymbol{u})] \le \mathbb{E}[c(\tau\boldsymbol{u})]. \tag{47}$$

Since $c(\tau\boldsymbol{u}) = c(\log\big(\boldsymbol{q}_0^{-1} \otimes \boldsymbol{q}_1\big)) = C(\boldsymbol{q}_0, \boldsymbol{q}_1)$ and $c(\tau'\boldsymbol{u}) = C(\boldsymbol{q}_0', \boldsymbol{q}_1')$. Therefore,

$$\mathbb{E}[C(\boldsymbol{q}_0', \boldsymbol{q}_1')] \le \mathbb{E}[C(\boldsymbol{q}_0, \boldsymbol{q}_1)]. \tag{48}$$

$\square$

## B. Implementation Details

### B.1. Ensuring The Shortest Geodesic Path on SO(3)

When we interpolate two quaternions by using SLERP in an exponential format (Eq. (8)), due to the double-cover property of quaternions (where every 3D rotation is represented by two antipodal unit quaternions), it is possible that the inner product $\langle \boldsymbol{q}_0, \boldsymbol{q}_1 \rangle < 0$, which means that $\boldsymbol{q}_0$ and $\boldsymbol{q}_1$ lie in opposite hemispheres. In such a situation, we apply $-\boldsymbol{q}_1$ in Eq. (8), ensuring the shortest geodesic path on SO(3).

### B.2. Auxiliary Loss

We adopt the auxiliary loss from (Yim et al., 2023b) to discourage physical violations such as chain breaks or steric clashes. Let $\mathcal{A} = [N, C_\alpha, C, O]$ be the collection of backbone atoms. The first term penalizes deviations in backbone atom coordinates:

$$\mathcal{L}_{\text{bb}} = \frac{1}{4N}\sum_{n=1}^{N}\sum_{a\in\mathcal{A}} \|a_n - \hat{a}_n\|^2, \tag{49}$$

where $a_n$ is the ground-truth atom position, $\hat{a}_n$ is our predicted position, $N$ represents the number of residues. The second loss is a local neighborhood loss on pairwise atomic distances,

$$\mathcal{L}_{\text{dis}} = \frac{1}{Z} \sum_{n,m=1}^{N} \sum_{a,b \in \mathcal{A}} \mathbf{1}\{d_{ab}^{nm} < 0.6\} \|d_{ab}^{nm} - \hat{d}_{ab}^{nm}\|^2, \tag{50}$$

$$Z = \left( \sum_{n,m=1}^{N} \sum_{a,b \in \mathcal{A}} \mathbf{1}\{d_{ab}^{nm} < 0.6\} \right) - N, \tag{51}$$

where $d_{ab}^{nm} = \|a_n - b_m\|$ and $\hat{d}_{ab}^{nm} = \|\hat{a}_n - \hat{b}_m\|$ represent true and predicted inter-atomic distances between atoms $a, b \in \mathcal{A}$ for residue $n$ and $m$. $\mathbf{1}$ is an indicator, signifying that only penalize atoms within 0.6nm(6Å). The full auxiliary loss can be written as

$$\mathcal{L}_{\text{aux}} = \mathcal{L}_{\text{bb}} + \mathcal{L}_{\text{dis}}. \tag{52}$$

### B.3. The Schemes of Training and Inference Algorithms

The schemes of our training and inference algorithms are shown below.

---

**Algorithm 1** Training Procedure of QFlow

---

**Require:** Training dataset $\mathrm{T}_1^{\mathcal{D}} = \left\{ \{\mathrm{T}_1^{\mathrm{j}} = (\mathrm{x}_1^{\mathrm{j}}, \mathrm{q}_1^{\mathrm{j}})\}_{j=1}^{N_i} \right\}_{i=1}^{D}$, model $\mathcal{M}_\theta$, number of epochs $N$

1: Initialize model parameters $\theta$
2: **for** epoch $= 1$ to $N$ **do**
3:    **for** each mini-batch $\mathrm{T}_1^{\mathcal{B}} \subset \mathrm{T}_1^{\mathcal{D}}$ **do**
4:       Sample $t^{\mathcal{B}} \sim \mathcal{U}[0, 1]$, $\mathrm{T}_0^{\mathcal{B}} \sim \mathcal{T}_0 \times \mathcal{Q}_0$
5:       Interpolate translations: $\boldsymbol{x}_t^{\mathcal{B}} = \text{Linear}(\boldsymbol{x}_0^{\mathcal{B}}, \boldsymbol{x}_1^{\mathcal{B}}, t^{\mathcal{B}})$                Eq. (7)
6:       Interpolate rotations: $\boldsymbol{q}_t^{\mathcal{B}} = \text{SLERP-Exp}(\boldsymbol{q}_0^{\mathcal{B}}, \boldsymbol{q}_1^{\mathcal{B}}, t^{\mathcal{B}})$             Eq. (8)
7:       Predict targets: $\boldsymbol{x}_{\theta,1}^{\mathcal{B}}, \boldsymbol{q}_{\theta,1}^{\mathcal{B}} = \mathcal{M}_\theta(\mathrm{T}_t^{\mathcal{B}}, t^{\mathcal{B}})$
8:       Compute loss $\mathcal{L}(\theta; \boldsymbol{x}_t^{\mathcal{B}}, \boldsymbol{q}_t^{\mathcal{B}}, \boldsymbol{x}_{\theta,1}^{\mathcal{B}}, \boldsymbol{q}_{\theta,1}^{\mathcal{B}}, t^{\mathcal{B}})$           Eq. (12)
9:       Compute gradient $\nabla_\theta \mathcal{L}$
10:     Update parameters: $\theta \leftarrow \theta - \eta \nabla_\theta \mathcal{L}$
11:    **end for**
12: **end for**
13: **Return:** Trained model parameters $\theta^*$

---

---

**Algorithm 2** Inference

---

**Require:** Trained model $\mathcal{M}_\theta$, noise $\mathrm{T}_0 \sim \mathcal{T}_0 \times \mathcal{Q}_0$, number of steps $L$, rotation acceleration constant $\gamma$

1: Initialize $t = 0$, $\Delta t = \frac{1}{L}$
2: **for** step $= 1$ to $L$ **do**
3:    Predict targets: $\boldsymbol{x}_{\theta,1}, \boldsymbol{q}_{\theta,1} = \mathcal{M}_\theta(\mathrm{T}_t, t)$
4:    Compute velocity: $\boldsymbol{v}_{\theta,t}, \boldsymbol{\omega}_{\theta,t}$                                    Eq. (10)
5:    Update translations: $\boldsymbol{x}_{t+\Delta t} \leftarrow \boldsymbol{x_t} + \boldsymbol{v}_{\theta,t} \cdot \Delta t$                Eq. (13)
6:    Update rotations: $\boldsymbol{q}_{t+\Delta t} \leftarrow \boldsymbol{q}_t \otimes \exp\left(\frac{1}{2}\Delta t \cdot \gamma e^{-\gamma t} \boldsymbol{\omega}_{\theta,t}\right)$     Eq. (15)
7:    Update states: $t \leftarrow t + \Delta t$, $\mathrm{T}_t \leftarrow \mathrm{T}_{t+\Delta t}$
8: **end for**
9: **Return:** Generated backbone frame $\mathrm{T}_1$

---

### B.4. Data Statistics and Hyperparameter Settings

We follow (Yim et al., 2023b) to construct PDB dataset. The dataset was downloaded on December 17, 2024. We then applied a length filter (60–512 residues) and a resolution filter ($< 5$ Å) to select high-quality structures. To further refine the

---

**Algorithm 3** Training Procedure of ReQFlow

---

**Require:** Trained QFlow model $\mathcal{M}_\theta$, number of epochs $N$
1: Sample noise $\mathrm{T}_0'^{\mathcal{D}} \sim \mathcal{T}_0 \times \mathcal{Q}_0$
2: Create flow rectification pairs: $(\mathrm{T}_0', \mathrm{T}_1')^{\mathcal{D}}$          Alg. 2
3: **for** epoch $= 1$ to $N$ **do**
4:     **for** each mini-batch $(\mathrm{T}_0', \mathrm{T}_1')^{\mathcal{B}} \subset (\mathrm{T}_0', \mathrm{T}_1')^{\mathcal{D}}$ **do**
5:        Sample $t^{\mathcal{B}} \sim \mathcal{U}[0,1]$
6:        Interpolate translations: $\boldsymbol{x}_t'^{\mathcal{B}} = \mathrm{Linear}(\boldsymbol{x}_0'^{\mathcal{B}}, \boldsymbol{x}_1'^{\mathcal{B}}, t^{\mathcal{B}})$       Eq. (7)
7:        Interpolate rotations: $\boldsymbol{q}_t'^{\mathcal{B}} = \mathrm{SLERP\text{-}Exp}(\boldsymbol{q}_0'^{\mathcal{B}}, \boldsymbol{q}_1'^{\mathcal{B}}, t^{\mathcal{B}})$       Eq. (8)
8:        Predict targets: $\boldsymbol{x}_{\theta,1}'^{\mathcal{B}}, \boldsymbol{q}_{\theta,1}'^{\mathcal{B}} = \mathcal{M}_\theta(\mathrm{T}_t'^{\mathcal{B}}, t^{\mathcal{B}})$
9:        Compute loss $\mathcal{L}(\theta; \boldsymbol{x}_t'^{\mathcal{B}}, \boldsymbol{q}_t'^{\mathcal{B}}, \boldsymbol{x}_{\theta,1}'^{\mathcal{B}}, \boldsymbol{q}_{\theta,1}'^{\mathcal{B}})$       Eq. (12)
10:       Compute gradient $\nabla_\theta \mathcal{L}$
11:       Update parameters: $\theta \leftarrow \theta - \eta \nabla_\theta \mathcal{L}$
12:     **end for**
13: **end for**
14: **Return:** Trained model parameters $\theta^*$

---

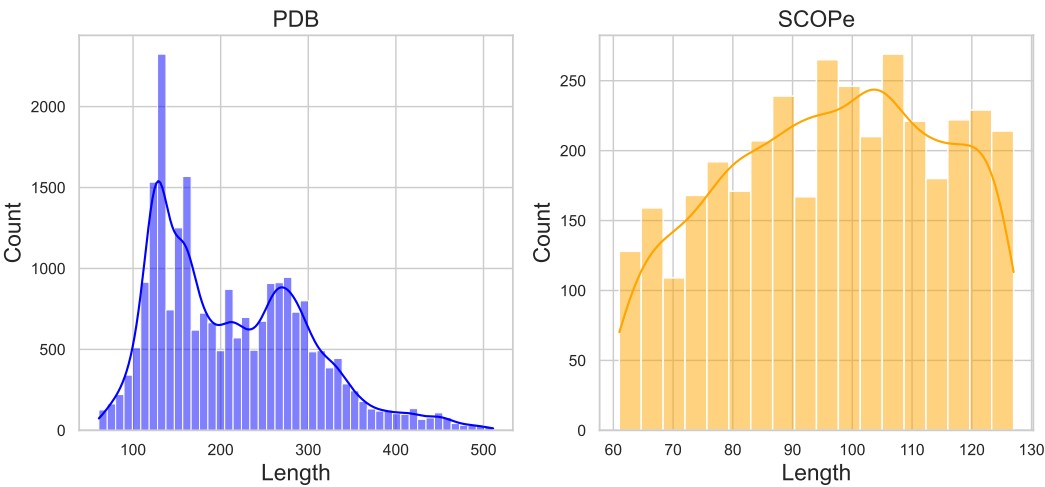

*Figure 6.* The length distribution of PDB and SCOPe dataset we use for training.

dataset, we processed each monomer using DSSP (Kabsch & Sander, 1983), removing those with more than 50% loops to ensure high secondary structure content. After filtering, 23,366 proteins remained for training. We directly use the SCOPe dataset preprocessed by (Yim et al., 2023a) for training, which consists of 3,673 proteins after filtering. The distribution of dataset length is shown on Figure 6.

When conducting reflow, we first generated a large amount of data to create the training dataset and then applied filtering to refine it. The filtering criteria were as follows: for proteins with lengths $\leq 400$, we selected samples with scRMSD $\leq 2$; for proteins with lengths $\geq 400$, we included samples with either scRMSD $\leq 2$ or TM-score $\geq 0.9$. We also remove those with more than 50% loop and those with max 4% radius gyration. For the PDB dataset, we generated 20 proteins for each length in $\{60, 61, \ldots, 512\}$, resulting in a reflow dataset containing 7,653 sample-noise pairs. For the SCOPe dataset, we generated 50 proteins for each length in $\{60, 61, \ldots, 128\}$, producing a reflow dataset with 3,167 sample-noise pairs.

### B.5. Metrics

Following existing work(Geffner et al., 2025; Yim et al., 2023b;a; Bose et al., 2024; Huguet et al., 2024), we apply the metrics below to evaluate various methods.

*Table 5.* Training Hyperparameters

| Hyperparameters | Value |
|---|---|
| aux_loss_t_pass (time threshold) | PDB=0.5, SCOPe=0.25 |
| aux_loss_weight | 1.0 |
| batch size | 128 |
| max_num_res_squared | PDB=1000000, SCOPe=500000 |
| max epochs | 1000 |
| learning rate | 0.0001 |
| interpolant_min_t | 0.01 |

**Designability**.We use this metric to evaluate whether a protein backbone can be formed by folding an amino acid chain. As shown in Figure 7, for each backbone, we generate 8 sequences with ProteinMPNN(Dauparas et al., 2022) at temperature 0.1, and predict their corresponding structures using ESMFold(Lin et al., 2023). Then we compute the minimum RMSD (known as scRMSD) between the predicted structures and the backbone sampled by the model. The designability score (denoted as "fraction" in this work) is the percentage of samples satisfying scRMSD $< 2$Å.

**Diversity**. This metric quantifies the diversity of the generated backbones. This involves calculating the average pairwise structural similarity among designable samples, broken down by protein length. Specifically, for each length $L$ under consideration, let $S_L$ be the set of designable structures. We compute $\text{TM}(b_i, b_j)$ for all distinct pairs $(b_i, b_j)$ within $S_L$. The mean of these TM-scores represents the diversity for length $L$. The final diversity score is the average of these means across all tested lengths $L$. Since TM-scores closer to 1 indicate higher similarity, superior diversity is reflected by lower values of this aggregated score.

**Novelty**. We evaluate the structural novelty by finding the maximum TM-score between a generated structure and any structure in the Protein Data Bank (PDB), using Foldseek(van Kempen et al., 2022). A lower resulting maximum TM-score signifies a more novel structure. The command(Geffner et al., 2025) utilized for this Foldseek search is configured as follows:

```
foldseek easy-search <pdb_path> <database> <aln_file> <tmp_folder>
--alignment-type 1 \
--exhaustive-search \
--max-seqs 10000000000 \
--tmscore-threshold 0.0 \
--format-output query,target,alntmscore,lddt,evalue
```

According to the issue of FoldSeek mentioned in https://github.com/steineggerlab/foldseek/issues/323, we use the E-value column to report the TM-score.

**Efficiency**. To ensure fairness, we measure inference time on idle GPU and CPU systems. For PDB-based models, we sampled 50 proteins of length 300 and reported the mean sampling time. Similarly, for SCOPe-based models, we sampled 50 proteins of length 128 and reported the mean sampling time. File saving and self-consistency calculations were excluded from the timing.

### B.6. Baselines

We compare our work with state-of-the-art methods in the community, including Genie2, RFdiffusion, FoldFlow/FoldFlow2, FrameFlow, and FrameDiff. We use the default checkpoints and parameters provided in these methods' repositories for our comparisons.

## C. More Experimental Details

### C.1. Hyperparameter Settings

We adopt the the same hyperparameter settings as FrameFlow for a fair comparison, and the key parameters are shown in Table 5.

*Table 6.* Computation time breakdown (in seconds) for different methods, datasets, and sampling steps. The time corresponds to generating a backbone with length $N = 128$ for SCOPe and $N = 300$ for PDB.

| Datasets | Methods | Steps | Model Prediction | Rotation Update | Translation Update | Total Time |
|---|---|---|---|---|---|---|
| PDB | FrameFlow | 500 | $16.308_{\pm0.093}$ | $0.608_{\pm0.005}$ | $0.033_{\pm0.000}$ | $17.053_{\pm0.099}$ |
| | | 50 | $1.609_{\pm0.013}$ | $0.059_{\pm0.001}$ | $0.003_{\pm0.000}$ | $1.727_{\pm0.014}$ |
| | | 20 | $0.635_{\pm0.008}$ | $0.024_{\pm0.001}$ | $0.001_{\pm0.000}$ | $0.713_{\pm0.010}$ |
| | QFlow | 500 | $16.732_{\pm0.089}$ | $0.492_{\pm0.004}$ | $0.036_{\pm0.000}$ | $17.370_{\pm0.111}$ |
| | | 50 | $1.670_{\pm0.003}$ | $0.048_{\pm0.000}$ | $0.003_{\pm0.000}$ | $1.776_{\pm0.004}$ |
| | | 20 | $0.653_{\pm0.001}$ | $0.019_{\pm0.000}$ | $0.001_{\pm0.000}$ | $0.726_{\pm0.002}$ |
| SCOPe | FrameFlow | 500 | $11.947_{\pm0.125}$ | $0.601_{\pm0.003}$ | $0.033_{\pm0.000}$ | $12.688_{\pm0.124}$ |
| | | 50 | $1.166_{\pm0.013}$ | $0.059_{\pm0.001}$ | $0.003_{\pm0.000}$ | $1.275_{\pm0.016}$ |
| | | 20 | $0.471_{\pm0.002}$ | $0.025_{\pm0.000}$ | $0.001_{\pm0.000}$ | $0.539_{\pm0.003}$ |
| | QFlow | 500 | $11.994_{\pm0.037}$ | $0.483_{\pm0.003}$ | $0.034_{\pm0.000}$ | $12.602_{\pm0.040}$ |
| | | 50 | $1.166_{\pm0.015}$ | $0.048_{\pm0.001}$ | $0.003_{\pm0.000}$ | $1.262_{\pm0.021}$ |
| | | 20 | $0.466_{\pm0.002}$ | $0.019_{\pm0.000}$ | $0.001_{\pm0.000}$ | $0.528_{\pm0.002}$ |

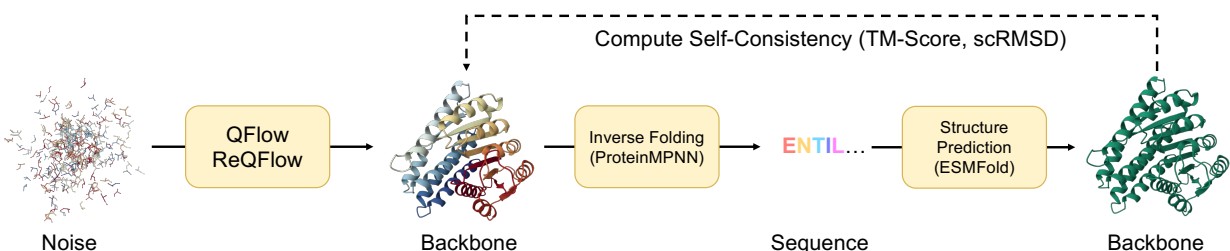

*Figure 7.* Illustration of the designability computation pipeline.

## C.2. Checkpoint selection strategy

For each method, after observing loss convergence, we select checkpoints based on the metrics of the generated protein validation set. We choose the checkpoint where the ca_ca_valid_percent $> 0.99$ and the proportions of secondary structures are closest to the dataset's average values.

## C.3. Detailed Speed Comparison

As shown in Table 6, we record the runtime (second) on generating a protein of length 300 in the PDB experiment and length 128 in the SCOPe experiment for a detailed speed comparison. The neural network feedforward computation is the main computational bottleneck. However, the quaternion operations are 15∼20% faster than rotation matrix-based operations (see the Rotation Update column).

## C.4. Detailed Comparisons Based on SCOPe

Table 7 reports the mean and standard deviation of the results corresponding to Table 4. The checkpoints of ReFrameFlow and ReQFlow used here are selected from epochs 2 to 6 of rectification training for fair comparison. The checkpoints of FrameFlow and QFlow used for rectification are from epoch 189 and epoch 195, respectively.

Table 8 presents comprehensive results from the SCOPe experiment using a fine-grained step size corresponding to Figure 5. Note that the chekponts of ReFrameFlow and ReQFlow here are different from Table 4. We select checkpoints following the criteria in Appendix C.2. Even with a generation process as concise as 10 steps, ReQFlow achieves a designable fraction of 0.848. This highlights the efficiency and effectiveness of ReQFlow in generating feasible protein structures. Additionally, both QFlow and ReQFlow models produce proteins with reasonable secondary structure distributions, indicating their capability to generate structurally plausible proteins. These findings underscore the potential of these models to significantly advance the field of protein design by balancing computational efficiency with structural accuracy.

*Table 7.* Comparisons for various models on SCOPe. For each metric of generation quality, we indicate the best and top-3 results in the same way as Table 2 does. The inference time corresponds to generating a backbone with length $N = 128$. For each method, we evaluate the results on five checkpoints and compute their mean and standard deviation.

| Method | Efficiency | | Designability | Diversity | Novelty |
|---|---|---|---|---|---|
| | Step | Time(s) | Fraction↑ | TM↓ | TM↓ |
| FrameFlow | 500 | 12.69 | $0.851_{\pm 0.016}$ | $0.392_{\pm 0.007}$ | $0.714_{\pm 0.003}$ |
| | 50 | 1.28 | $0.811_{\pm 0.017}$ | $\mathbf{0.378_{\pm 0.008}}$ | $0.682_{\pm 0.004}$ |
| | 20 | 0.54 | $0.708_{\pm 0.023}$ | $0.370_{\pm 0.005}$ | $0.649_{\pm 0.008}$ |
| ReFrameFlow | 500 | 12.77 | $0.924_{\pm 0.007}$ | $0.407_{\pm 0.004}$ | $0.709_{\pm 0.004}$ |
| | 50 | 1.26 | $0.906_{\pm 0.006}$ | $0.407_{\pm 0.001}$ | $0.689_{\pm 0.003}$ |
| | 20 | 0.52 | $0.884_{\pm 0.009}$ | $0.405_{\pm 0.004}$ | $0.714_{\pm 0.003}$ |
| QFlow | 500 | 12.60 | $0.893_{\pm 0.022}$ | $0.392_{\pm 0.005}$ | $0.707_{\pm 0.011}$ |
| | 50 | 1.26 | $0.854_{\pm 0.019}$ | $0.380_{\pm 0.005}$ | $0.675_{\pm 0.014}$ |
| | 20 | 0.53 | $0.762_{\pm 0.015}$ | $0.372_{\pm 0.004}$ | $0.644_{\pm 0.013}$ |
| ReQFlow | 500 | 12.52 | $\mathbf{0.947_{\pm 0.007}}$ | $0.406_{\pm 0.003}$ | $0.696_{\pm 0.003}$ |
| | 50 | 1.30 | $0.922_{\pm 0.007}$ | $0.411_{\pm 0.002}$ | $0.680_{\pm 0.007}$ |
| | 20 | 0.53 | $0.910_{\pm 0.012}$ | $0.405_{\pm 0.001}$ | $\mathbf{0.671_{\pm 0.005}}$ |

## C.5. Comparisons on Model Size and Training Data Size

The comparison of model size and training dataset size is listed in Table 9. Model sizes in the table refer to the number of *total* parameter. FoldFlow2 utilizes a pre-trained model, thus having 672M parameters in total. The number of trainable parameters is 21M.

## C.6. Visualization Results

We use *Mol Viewer* (Sehnal et al., 2021) to visualize protein structures generated by different models, as shown in Figure 8 and Figure 9. In Figure 8, all proteins originate from the same noise initialization generated by QFlow, whereas in Figure 9, the initialization is generated by FoldFlow. Each method follows its own denoising trajectory, leading to distinct structural outputs. FoldFlow2 adopts a default sampling step of 50, while all other methods use 500 steps. Due to architectural differences, the final structures vary across models, but within the same model, different sampling steps generally yield similar structures.

Among all models, ReQFlow exhibits the most stable and robust performance, maintaining low RMSD and variance across different sampling steps while demonstrating resilience to varying noise inputs. In contrast, other methods show significant limitations. FoldFlow-OT is highly sensitive to initial noise, displaying drastically different performance in Figure 8 and Figure 9—evidenced by substantial variance across sampling steps when using QFlow noise.

Moreover, FoldFlow-OT tends to overproduce $\alpha$-helices—coiled, spiral-like structures—resulting in high designability scores but deviating from realistic protein distributions. This pattern suggests a high risk of mode collapse, where the model predominantly learns a specific subset of protein structures, thereby lacking diversity and novelty in its predictions. Conversely, ReQFlow and QFlow generate a higher proportion of $\beta$-strands, which appear as extended, ribbon-like structures, indicating a closer alignment with natural protein distributions.

Furthermore, as sampling steps decrease, most baseline models experience a sharp deterioration in quality: RMSD values increase, rendering the structures non-designable. In extreme cases, some samples exhibit severe fragmentation or disconnected backbones (e.g., the dashed regions in FoldFlow2 at 20 steps, Figure 8), highlighting instabilities in their sampling dynamics.

*Table 8.* Unconditional protein backbone generation performance for 10 samples each length in $\{60, 61, \cdots, 128\}$. We report the metrics from Section 4.1 and we indicate the best and top-3 results in the same way as Table 2 does.

| | Step | Designability | | Diversity | Novelty | Sec. Struct. | |
|---|---|---|---|---|---|---|---|
| | | Fraction ($\uparrow$) | scRMSD ($\downarrow$) | TM ($\downarrow$) | TM ($\downarrow$) | Helix | Strand |
| Scope Dataset | - | - | - | - | - | 0.330 | 0.260 |
| FrameFlow | 500 | 0.849 | $1.448_{\pm 1.114}$ | 0.397 | 0.713 | 0.439 | 0.236 |
| | 400 | 0.864 | $1.353_{\pm 0.890}$ | 0.390 | 0.713 | 0.452 | 0.229 |
| | 300 | 0.861 | $1.422_{\pm 1.178}$ | 0.389 | 0.715 | 0.449 | 0.230 |
| | 200 | 0.842 | $1.496_{\pm 1.411}$ | 0.387 | 0.704 | 0.437 | 0.237 |
| | 100 | 0.823 | $1.517_{\pm 1.228}$ | 0.388 | 0.697 | 0.426 | 0.238 |
| | 50 | 0.820 | $1.546_{\pm 1.316}$ | 0.379 | 0.685 | 0.441 | 0.228 |
| | 20 | 0.713 | $1.918_{\pm 1.495}$ | 0.362 | 0.656 | 0.416 | 0.219 |
| | 10 | 0.504 | $2.924_{\pm 2.362}$ | 0.381 | 0.626 | 0.363 | 0.213 |
| ReFrameFlow | 500 | 0.897 | $1.368_{\pm 1.412}$ | 0.403 | 0.700 | 0.501 | 0.187 |
| | 400 | 0.893 | $1.328_{\pm 0.763}$ | 0.405 | 0.698 | 0.489 | 0.202 |
| | 300 | 0.888 | $1.313_{\pm 0.686}$ | 0.405 | 0.697 | 0.485 | 0.199 |
| | 200 | 0.907 | $1.326_{\pm 0.761}$ | 0.408 | 0.689 | 0.482 | 0.206 |
| | 100 | 0.886 | $1.322_{\pm 0.804}$ | 0.410 | 0.690 | 0.499 | 0.201 |
| | 50 | 0.903 | $1.291_{\pm 0.763}$ | 0.404 | 0.685 | 0.504 | 0.202 |
| | 20 | 0.871 | $1.416_{\pm 0.880}$ | 0.406 | 0.675 | 0.528 | 0.190 |
| | 10 | 0.806 | $1.696_{\pm 1.093}$ | 0.390 | **0.650** | 0.496 | 0.192 |
| QFlow | 500 | 0.907 | $1.263_{\pm 1.334}$ | 0.389 | 0.712 | 0.498 | 0.214 |
| | 400 | 0.907 | $1.199_{\pm 0.847}$ | 0.394 | 0.711 | 0.476 | 0.223 |
| | 300 | 0.910 | $1.243_{\pm 1.027}$ | 0.393 | 0.710 | 0.503 | 0.209 |
| | 200 | 0.877 | $1.309_{\pm 1.208}$ | 0.389 | 0.714 | 0.481 | 0.224 |
| | 100 | 0.903 | $1.283_{\pm 1.027}$ | 0.390 | 0.702 | 0.476 | 0.225 |
| | 50 | 0.872 | $1.389_{\pm 1.314}$ | **0.371** | 0.674 | 0.491 | 0.206 |
| | 20 | 0.764 | $1.764_{\pm 1.529}$ | 0.367 | 0.646 | 0.492 | 0.192 |
| | 10 | 0.565 | $2.589_{\pm 2.216}$ | 0.374 | 0.614 | 0.467 | 0.167 |
| ReQFlow | 500 | **0.972** | $\mathbf{1.043}_{\pm 0.416}$ | 0.418 | 0.703 | 0.507 | 0.228 |
| | 400 | 0.962 | $1.050_{\pm 0.445}$ | 0.417 | 0.697 | 0.523 | 0.212 |
| | 300 | 0.962 | $1.076_{\pm 0.518}$ | 0.421 | 0.702 | 0.498 | 0.233 |
| | 200 | 0.948 | $1.084_{\pm 0.509}$ | 0.407 | 0.696 | 0.513 | 0.218 |
| | 100 | 0.933 | $1.123_{\pm 0.669}$ | 0.425 | 0.695 | 0.514 | 0.310 |
| | 50 | 0.932 | $1.162_{\pm 0.812}$ | 0.422 | 0.693 | 0.491 | 0.237 |
| | 20 | 0.929 | $1.214_{\pm 0.633}$ | 0.409 | 0.670 | 0.514 | 0.307 |
| | 10 | 0.848 | $1.546_{\pm 0.944}$ | 0.416 | 0.662 | 0.518 | 0.195 |

*Table 9.* Model Sizes and Training Dataset Sizes

| Model | Training Dataset Size | Model Size (M) |
|---|---|---|
| RFDiffusion | >208K | 59.8 |
| Genie2 | 590K | 15.7 |
| FrameDiff | 23K | 16.7 |
| FoldFlow(Base,OT,SFM) | 23K | 17.5 |
| FoldFlow2 | ~160K | 672 |
| FrameFlow | 23K | 16.7 |
| QFlow | 23K | 16.7 |
| ReQFlow | 23K+7K | 16.7 |

[1] When training ReQFlow, we first apply the 23K samples of PDB to train QFlow, and then we use additional 7K samples generated by QFlow in the flow rectification phase.

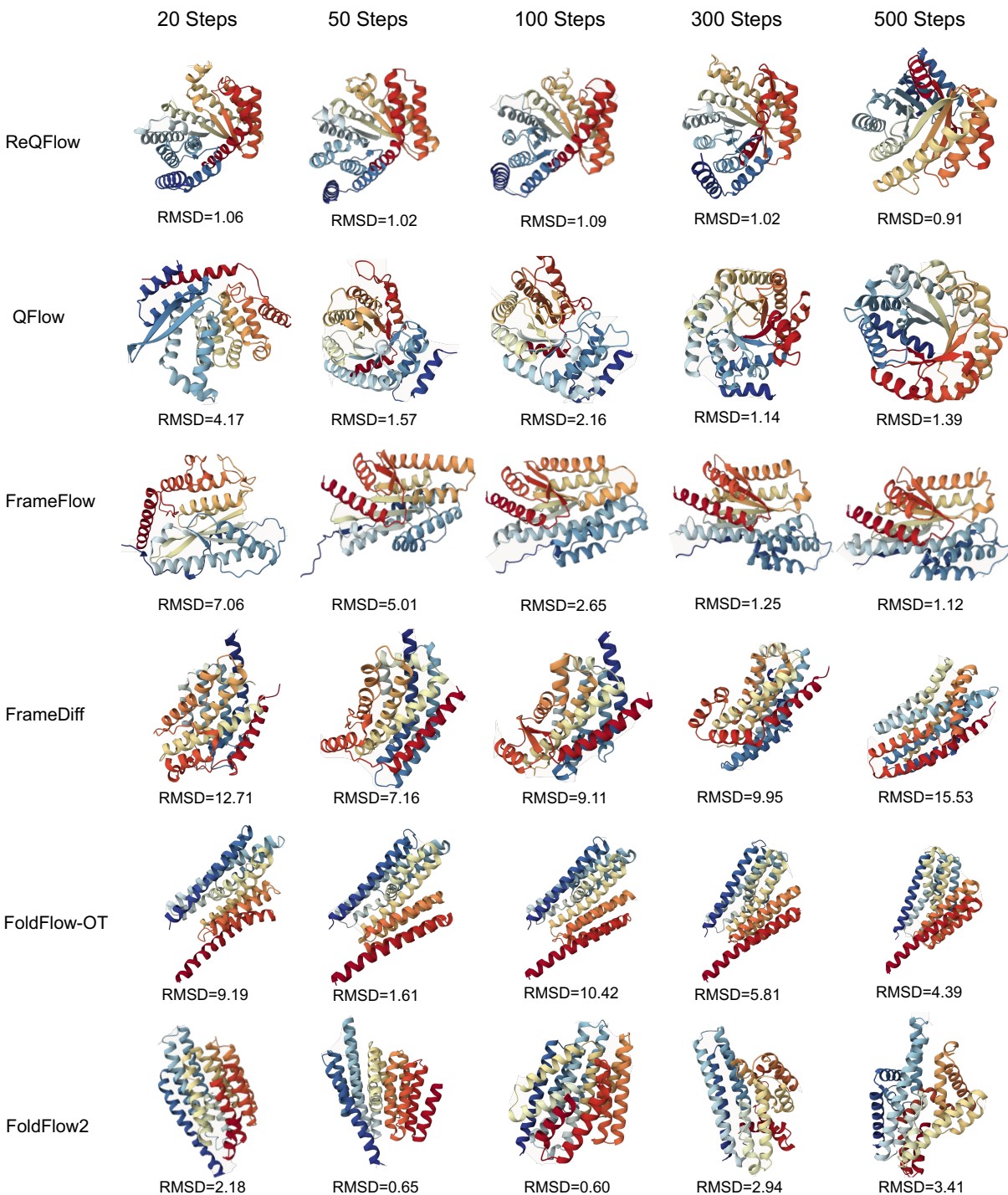

*Figure 8.* Visualization of different methods on length 300. Sampling start with a *same* noise generated by QFlow.

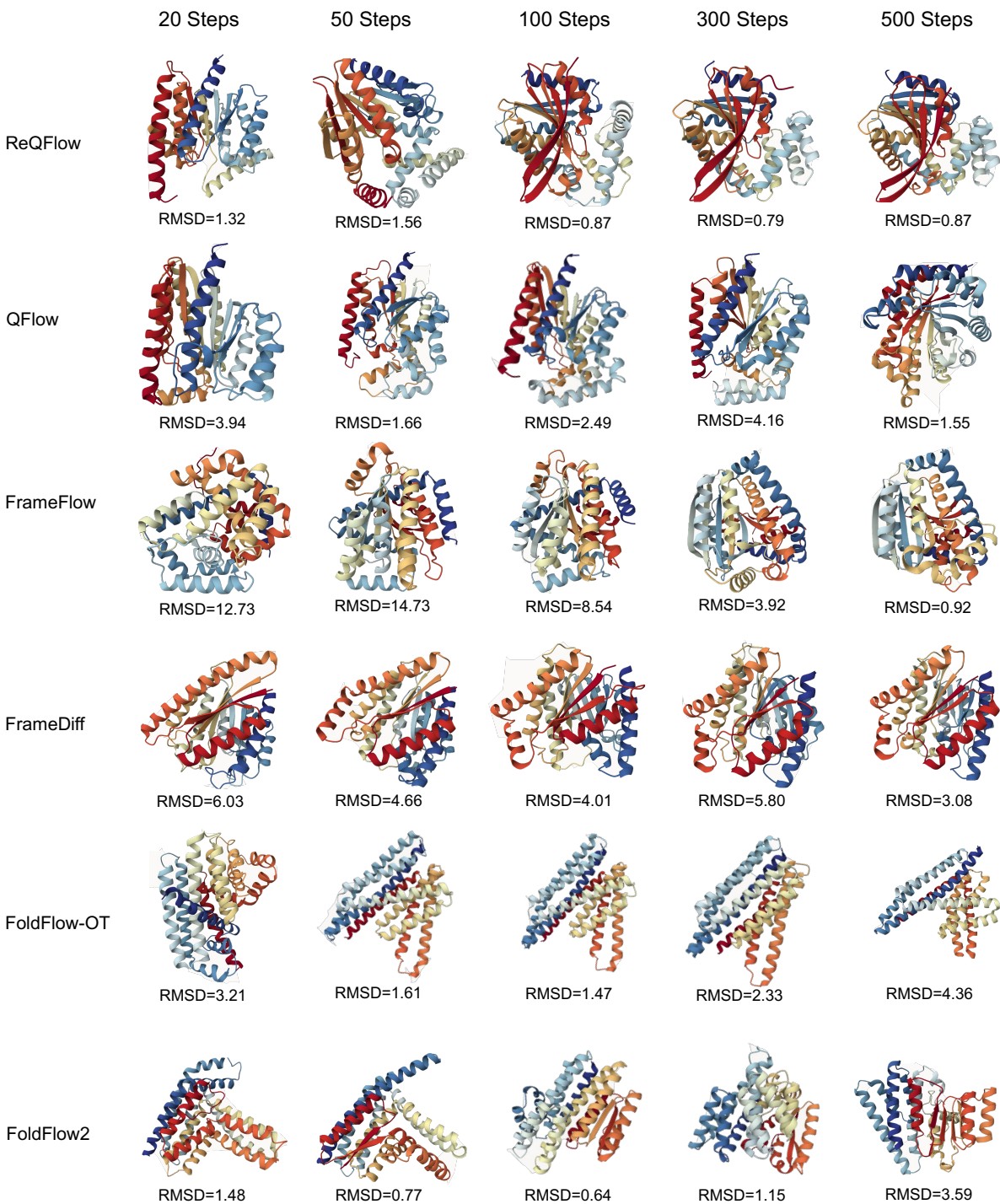

*Figure 9.* Visualization of different methods on length 300. Sampling start with a *same* noise generated by FoldFlow.

*Table 10.* Comparisions for various methods on their performance (Fraction Score) in long backbone generation. The lengths of the generated backbones range from 300 to 600. We generate 50 samples for each length. We bold the best result and show the top-3 results with a blue background.

| Length $N$ | 300 | 350 | 400 | 450 | 500 | 550 | 600 |
|---|---|---|---|---|---|---|---|
| RFDiffusion | 0.76 | 0.70 | 0.46 | 0.36 | 0.20 | 0.20 | 0.10 |
| Genie2 | 0.86 | 0.90 | 0.74 | 0.58 | 0.28 | 0.12 | 0.10 |
| FoldFlow2 | 0.96 | 0.88 | 0.70 | 0.56 | 0.60 | 0.26 | **0.16** |
| FrameDiff | 0.24 | 0.18 | 0.00 | 0.00 | 0.00 | 0.00 | 0.00 |
| FoldFlow-OT | 0.62 | 0.48 | 0.30 | 0.10 | 0.04 | 0.00 | 0.00 |
| FrameFlow | 0.72 | 0.74 | 0.48 | 0.28 | 0.24 | 0.10 | 0.00 |
| QFlow | 0.88 | 0.78 | 0.54 | 0.50 | 0.30 | 0.02 | 0.00 |
| ReQFlow | **0.98** | **0.96** | **0.78** | **0.76** | **0.70** | **0.56** | 0.10 |

