# OpenReview forum: "ReQFlow: Rectified Quaternion Flow for Efficient and High-Quality Protein Backbone Generation"
_ICML.cc/2025/Conference — ICML 2025 poster_

### Official Review · Reviewer_oNdZ · 2025-03-11

**Overall Recommendation:** 2

**Summary:**

The paper introduces ​ReQFlow, a novel method for protein backbone generation. To ​address numerical instability in matrix-based representations, the main innovation is to use quaternions to model the rotations and spherical linear interpolation in the flow matching training. The paper also extends the rectified flow techniques to quaternion space. The resulting model achieves more than 30x speedup over RFDiffusion/Genie2 while maintaining high designability.

**Claims And Evidence:**

Yes

**Essential References Not Discussed:**

NA

**Experimental Designs Or Analyses:**

Yes.

**Methods And Evaluation Criteria:**

Yes

**Other Comments Or Suggestions:**

NA

**Other Strengths And Weaknesses:**

1. The theoretical part of the paper is substantial, but the innovation is a bit limited.
2. Some parts of the experiments are not clear. e.g., when to stop training?
3. The training dataset can be improved. PDB data biases the model toward naturally abundant folds. Or sampling strategies to ensure structure diversity can be applied.

**Questions For Authors:**

NA

**Relation To Broader Scientific Literature:**

ReQFlow builds upon and advances several prior works in protein generation, flow-based models, and quaternion representations.

**Theoretical Claims:**

Yes, for theorems 3.1 and 3.2, they are very similar to the results in the original rectified flow paper (https://arxiv.org/abs/2209.03003 theorems 3.3 and 3.5).

---

> ### Author Rebuttal · Authors · 2025-04-01
>
> Thanks for your comments.
>
> **Correction of long-chain results:** Before resolving your concerns, we would like to report that we found a bug in our script in the rebuttal phase and correct the results of ReQFlow on long-chain generation in the anonymous link https://anonymous.4open.science/r/6342_ReQFlow_Rebuttal-62BB/01_revised_longchain.pdf. **Note that, the new results are lower than the wrong ones in the submission, but they are still significantly better than those of baselines, and thus, do not affect our main claims and contributions.**
>
> Below we try to resolve your concerns one by one:
>
> **Q1. The novelty of our work, especially the theoretical part.**
>
> **A1.** As we claimed in the supplementary file, our theoretical result is a natural extension of Rectified Flow [1] and its proof applies the same pipeline. However, we indeed makes the first attempt to extend the theoretical result in [1] to the SO(3) scenario. In addition, technically, our approach is novel for the following reasons:
>
> First, we innovatively replace rotation matrices with quaternions. Thanks to the quaternion representations, our QFlow significantly enhances designability compared to its counterpart FrameFlow, without sacrificing diversity and novelty (see Tables 2 and 4). Moreover, QFlow exhibits superior efficiency, achieving approximately 10% and 25% speedups on the PDB and SCOPe datasets, respectively (see lines 358-370, Tables 2 and 4).
>
> Second, although the rectified flow technique was proposed in [1], we pioneeringly apply it to the protein generation and provide theoretical guarantees on SO(3). As demonstrated in the table below, ReQFlow markedly improves efficiency while maintaining the other three metrics on par with other advanced models.
>
> ||training set|model size|steps|time (s)|designability|diversity|novelty|
> |-|-|-|-|-|-|-|-|
> |Genie2|590k|15.7|1000|112.93|0.908|0.370|0.759|
> |RFDiffusion|>208k|59.8|50|66.23|0.904|0.382|0.822|
> |FoldFlow2|160k|672|50|6.35|0.952|0.373|0.813|
> |ReQFlow|30k|16.7|50|1.81|0.912|0.369|0.810|
>
> [1] Flow Straight and Fast: Learning to Generate and Transfer Data with Rectified Flow
>
> **Q2. More experimental details**
>
> **A2.** We adopt the the same hyperparameter settings as FrameFlow for a fair comparison, and the key parameters are shown below:
> |Hyperparameters|Values|
> |-|-|
> |aux\_loss\_t\_pass (time threshold)|PDB=0.5, SCOPe=0.25|
> |aux\_loss\_weight|1.0|
> |batch size|128|
> |max\_num\_res\_squared|PDB=1000000, SCOPe=500000|
> |max epochs|1000|
> |learning rate|0.0001|
> |interpolant\_min\_t|0.01|
>
> Note that the batch size of FrameFlow is originally set to 128 for PDB and 100 for SCOPe. We retrained FrameFlow on SCOPe in our work and set the batch size to 128, the same as QFlow/ReQFlow. This doesn't influence the comparison's effectiveness and fairness.
>
> As for when to stop training, we found that the loss converges after 600 epochs on the PDB dataset and 200 epochs on the SCOPe dataset.
>
> **Q3. Improve the diversity of training dataset by sampling**
>
> **A3.** For a fair comparison, we set our training dataset and training pipeline to be the same with those of the baseline models (e.g., FrameFlow, FoldFlow, and FrameDiff), so that the performance of various methods is achieved under the same setting and comparable. To our knowledge, existing methods, including the baselines mentioned above, train their models on the raw PDB dataset. Because the experimental setting is same, so the superiority of our method can be attributed to our technical contributions clearly.
>
> In addition, as shown in Figure 3, the distribution of generated proteins obtained by our method is comparable to that of PDB, which does not suffer severe mode collapse issue as FoldFlow does. The diversity and novelty scores shown in Tables 2 and 4 also indicate that the proteins generated by our method have reasonable diversity and novelty.
>
> Sampling PDB data may mitigate the inductive bias in the dataset and further improve our model performance, but it is out of the scope of our work at the current stage.
>
> In summary, we hope the above responses can resolve your concerns and help you re-evaluate our work. We would appreciate it if you consider raising your score based on our reply. Feel free to contact us if you have any other questions.

---

### Official Review · Reviewer_K7Q3 · 2025-03-13

**Overall Recommendation:** 3

**Summary:**

This paper focuses on the task of protein backbone generation. It proposes quaternion flow (QFlow) and rectified quaternion flow (ReQFlow) for generative modeling on a translation/rotation manifold. In particular, in contrast to previous work, QFlow models the rotations with quaternions and the authors introduce quaternion flow matching. In contrast to previous works, the quaternion formulation avoids numerical instabilities and is an elegant way to describe rotations. The authors then use the rectified flow method to rectify and accelerate their models, making it possible to sample with relatively few steps. The paper then applies the method to protein backbone generation, where each protein residue is represented by its backbone coordinate and a rotation on SO(3). The paper generally achieves appealing results, slightly outperforming or being on-par with previous methods on standard benchmarks and evaluation metrics.

**Claims And Evidence:**

The authors claim that the proposed method, i.e. the quaternion flow matching and rectification, improves protein designability, accelerates generation, and overall achieves state-of-the-art performance. While the proposed method generally shows good performance, I think the "state-of-the-art" comment might be exaggerated.

- Generally, it is well-known that there is a trade-off between designability and diversity and novelty, and different methods achieve different trade-offs along this Pareto frontier. The state-of-the-art claim would be appropriate if the method clearly achieved both the best designability *and* diversity/novelty in the same setting, but this is not the case. After all, in theory, we can achieve 100% designability trivially by always outputting the same designable protein, in the extreme case.
  - We see that QFlow gets best diversity in Table 2, but not designability.
  - ReQFlow gets very good designability (in part potentially due to designable protein filtering, see comment below), but shows reduced diversity. A similar trend is also visible in Table 3.
  - Hence, it would be more appropriate to claim that the method performs "on-par" with existing works.
- Moreover, ReQFlow, in fact, not only does rectification but also blends in a distillation procedure where the rectification is only carried out on designable high-quality proteins. The fact that ReQFlow without this does not work well is concerning. Also, none of the baselines is trained on 100% designable protein backbones, but these baselines would potentially benefit from training on purely designable proteins, too. This makes the comparisons and claims somewhat questionable.

**Essential References Not Discussed:**

I cannot identify any essential references that are not discussed.

**Experimental Designs Or Analyses:**

I do not have any concerns regarding the general experimental design or analyses and no reason to believe that any experiment should be incorrect. My concerns are described above.

**Methods And Evaluation Criteria:**

In general, all proposed methods and evaluation criteria do make sense. My more specific concerns are outlined above.

**Other Comments Or Suggestions:**

I do not have any further comments or suggestions.

**Other Strengths And Weaknesses:**

**Strengths:**
The paper is generally well-written and well-presented and easy to read. The proposed quaternion flow matching is novel and original, and successfully validated. I think the proposed quaternion approach makes sense to model rotations in this setting. As the authors pointed out, similar methods have been used elsewhere in other domains.

**Weaknesses:**
- As mentioned above, I think the paper is a bit "overclaiming" when saying it gets state-of-the-art results.
- As discussed above, the ReQFlow method suffers from reduced diversity. There are trade-offs between designability and diversity.
- The paper focuses on simple protein backbone generation and, as discussed, performs approximately on-par with similar work overall, or slightly better when looking at individual metrics like designability only. In practical settings, protein generation is almost always carried out in a conditional setting. For instance, there is a target protein, for which we want to generate a binding protein, or there is a motif, and we want to generate the scaffold. Unfortunately, the paper does not study any such tasks.

Overall, I think the proposed method can be broadly useful to the community for modeling proteins with frame-based representations, including rotations on SO(3). While I do have criticisms, I do not see any fundamental flaws and hence I am generally leaning towards suggesting acceptance. I would be willing to raise my score for a rebuttal addressing my questions.

**Questions For Authors:**

1. The exponential step size scheduler for generating the rotations plays a critical role. This is well-known and similar to previous works. Nonetheless, why exactly do you think that this is the case?

2. Related, did you train the model with one joint interpolation time $t$ for translations and rotations, or did you sample separate independent $t$ for rotations and translations during training?

3. If you used one joint interpolation time $t$, did you consider not only running inference with the accelerated rotation schedule but to also train the model with the accelerated schedule (accelerated relative to the translations)?

**Relation To Broader Scientific Literature:**

The paper's key contributions are appropriately positioned with respect to the broader literature. In particular, previous works that conduct flow matching on rotation manifolds are cited and discussed, and works that leverage similar quaternion formulations in other machine learning areas are also cited and discussed. The most relevant related works in protein generative modeling are also discussed and cited.

**Theoretical Claims:**

The theoretical claims and proofs in the appendix all seem correct to me. These are mostly generalizations of the derivations done in the rectified flow paper to SO(3).

---

> ### Author Rebuttal · Authors · 2025-04-01
>
> Thanks for your positive feedback and constructive comments.
>
> **Correction of long-chain results:** Before resolving your concerns, we would like to report that we found a bug in our script in the rebuttal phase and correct the results of ReQFlow on long-chain generation are in https://anonymous.4open.science/r/6342_ReQFlow_Rebuttal-62BB/01_revised_longchain.pdf. **Note that, the new results are lower than the wrong ones in the submission, but they are still significantly better than those of baselines on designability and efficiency and comparable on novelty and diversity. This correction does not affect our main claims and contributions.**
>
> Below we try to resolve your concerns one by one:
>
> **Q1. Trade-off among different metrics and the definition of SOTA performance**
>
> **A1.** In our work, the "SOTA performance" means that **ReQFlow achieves higher designability and computational efficiency than the baselines and obtains comparable diversity and novelty**. We emphasize on designability because novelty and diversity are computed based on designable proteins. We can obtain very low TM-scores even if the protein backbones are random noise, which, however, is meaningless.
>
> To be clear, it does not mean that we do not care about novelty and diversity. We have offered experimental results for all metrics in Tables 2 and 4. In addition, although the diversity score degrades slightly after rectifying QFlow, the data distribution of ReQFlow in Figure 3 is similar to that of PDB. It means that ReQFlow has a low risk of mode collapse and achieves reasonable diversity.
>
> We will claim the trade-off among the metrics and eliminate any potential misleading content in the revised paper.
>
> **Q2. The significance of Reflow and designable samples**
>
> **A2.** In our opinion, both Reflow and designable samples are important.
>
> Applying Reflow without selecting designable samples means fine-tuning the model under the supervision of noisy samples, which naturally leads to performance degradation (see Table 3). **However, it does not mean that Reflow is useless.** To verify our claim, we fine-tune a trained QFlow on generated designable samples, leading to "self-distill QFlow". The comparison between the self-distill QFlow and ReqFlow is in https://anonymous.4open.science/r/6342_ReQFlow_Rebuttal-62BB/02_reflow_vs_selfdistill.pdf. Although self-distill QFlow obtains competitive results (comparable designability, slightly worse novelty and diversity), it changes the data distribution and suffers a high risk of mode collapse.
>
> In summary, our contributions include 1) first introducing quaternion algebra into flow-matching and protein design; and 2) making the first attempt to apply Reflow to protein design. For the second contribution, it is a potential technical route. Even if it is not consistently better than self-distillation, it does not mean it is not worth to explore.
>
> Self-distillation v.s. ReFlow, which one is better? It is interesting and can be our future work, but it is out of the scope of this work.
>
> **Q3. Conditional generation.**
>
> **A3.** Existing methods (FrameFlow, FoldFlow, etc.) focus on unconditioned backbone generation and design experiments accordingly. We just follow the same setting for a fair comparison.
>
> Nevertheless, we follow your suggestion, training QFlow and FrameFlow on SCOPe like [1] did. We select three motifs (4JHW, 5IUS and 1PRW) and report the success rate (%) of generation (i.e., scRMSD≤2A, motifRMSD≤1A). The results show that QFlow is at least comparable to FrameFlow.
>
> ||4JHW|5IUS|1PRW|
> |-|-|-|-|
> |FrameFlow|4|76|99|
> |QFlow|8|76|99|
>
> [1] Improved motif-scaffolding with SE(3) flow matching
>
> **Q4. Why does exponential step scheduler matter?**
>
> **A4.** Given a trained model, at the early stage of inference, i.e., $t\in [0, 0.25]$, the loss is significantly higher than that near the endpoint (i.e., $t\in [0.75, 1]$), as shown in https://anonymous.4open.science/r/6342_ReQFlow_Rebuttal-62BB/rot_loss.pdf. It means that **the vector field is not well-learned in the early phase.** The reason for this phenomenon might be **the models like FoldFlow and FrameFlow learn the velocity field backward.** This strategy makes the predictions far from the endpoint challenging.
>
> Exponential scheduler allows the model to rapidly approach the endpoint with few steps, reducing error accumulation caused by the imprecise samples at the early stage.
>
> **Q5. Joint interpolation time? and acceleration in training.**
>
> **A5.** Yes, we train the model with one joint interpolation time. As noted in the Eq.(7) of the FrameFlow paper, they attempted to accelerate the training phase as well, but this made training too easy with little learning happening. This aligns with our observations in **A4**: accelerating training further exacerbates the difficulty of learning the velocity field at the starting point.
>
> In summary, we hope the above responses can resolve your concerns and help enhance your confidence to raise your score.

---

### Official Review · Reviewer_TTp8 · 2025-03-19

**Overall Recommendation:** 3

**Summary:**

The paper proposes a method to train a generative model of protein backbones. They follow previous work in parameterizing protein backbones using a translation and a rotation. They have two main contributions:

- The use of normalized quaternions to parameterize rotations (most previous work use 3x3 rotation matrices)

- The application of the ReFlow process to enable fast sampling through additional training on synthetic data. The paper also extends the theoretical results from ReFlow (diminishes transport cost).

The empirical evaluation shows promising results for the proposed method.

**Claims And Evidence:**

Partly. As I state in the sections below, I think a more careful discussion of the results (including all metrics), would be useful, and an extended analysis of the numerical stability of using unit quaternions (vs 3x3 rotation matrices) would strengthen the paper.

**Essential References Not Discussed:**

Most relevant references are included. However, please see the section “Other comments and suggestions” for some issues connected to an existing paper.

**Ethical Review Concerns:**

As mentioned in “Other Comments Or Suggestions”, sections B.5 and B.6 in the Appendix appear to be heavily copied from another paper (which is not cited nor mentioned). To be clear, this does not affect the paper contributions, as those sections mostly describe baselines and metrics. However, copy/pasting content from another paper, without citing the source, does not seem to the correct thing to do.

**Ethical Review Flag:**

Flag this paper for an ethics review.

**Ethics Expertise Needed:**

["Other expertise"]

**Experimental Designs Or Analyses:**

As mentioned above, the metrics reported in the tables are very reasonable and widely used in the literature. The authors, however, appear to heavily rely on the designability metric for most of its analyses and claims in the text, which could be problematic. Most methods (including, for instance, Genie2) are able to achieve very high designabilities by changing the temperature parameter during sampling. However, there’s usually a tradeoff between designability and diversity, and reducing sampling temperature often leads to reduced diversity. While the tables include several metrics (designability, diversity, novelty), it would be good for the text to discuss the results for all metrics more. For instance, line 375 (left column) claims that ReQFlow outperforms Genie2 and other baselines, but the comparison is only done in terms of designability. Taking the same lines in the table referred to in the text, it can be seen that ReQFlow, for that configuration, achieves a worse diversity and novelty than Genie2. Having a more comprehensive analysis, commenting on these tradeoffs, and all metrics, would be good.

Another example, table 7 in the appendix only includes designability. Adding other metrics to get a more holistic view of the methods’ performance would be informative.

**Methods And Evaluation Criteria:**

The evaluation criteria follows standard practices in unconditional protein backbone design, training on a few subsets of the PDB, and evaluating widely used metrics such as designability, diversity, novelty, and secondary structure content.

**Other Comments Or Suggestions:**

There is some discrepancy between the description of diversity in the main paper and in the appendix. The main paper states that the pairwise tm-scores are averaged, while the appendix states that this is done per length, and then averaged across lengths. Which one is used for the results shown in the paper?

Additionally, Sections B.5 and B.6 in the Appendix appear to be copied from the paper “Proteina: Scaling Flow-based Protein Structure Generative Models”. While there are small changes (a few words here and there), there are entire paragraphs that are almost an exact copy. Since this is just describing metrics and baselines, and not a core part of the method, I don’t consider this to be very problematic. But copy / pasting entire paragraphs, especially without citing / mentioning the source (!) (the original paper, which also deals with backbone design using flow matching, is not even mentioned), seems in general unacceptable, even more so for an ICML submission.

On this line, the Genie2 description in B.6 states that the noise scale was set to 1 for full temperature sampling. However, this full temperature sampling is not included in any of the results shown in the paper.

**Other Strengths And Weaknesses:**

One weakness that came up when looking at the results in some detail, about the application of ReFlow. Using ReFlow requires creating a synthetic dataset, where each sample is obtained by fully simulating a pre-trained flow model.

(w1) The synthetic datasets created are filtered to keep only designable samples. Without this filter performance of the ReFlow-ed model takes a bit of a hit. Can the authors explain why this may be the case? Under perfect ReFlow marginals should be preserved? Is the important component here the ReFlow or the designability filter?

(w2) Generating the ReFlow synthetic dataset is quite expensive. In fact, the datasets created in the paper are somewhat small, consisting of roughly 5k samples. After ReFlow, designability goes up (this is likelt thanks to the self-distillation effect achieved by filtering the synthetic data for only designable samples), but diversity and novelty get worse. The authors don’t really comment on this. Could this be addressed by generating a larger synthetic dataset?

**Questions For Authors:**

Reading other papers, such as FrameFlow and FoldFlow, I was always curious about the gamma parameter to control the rotation generation speed. Do you have anh intuitions why this works so well? Most methods, including the one presented in the paper, perform quite poorly without it.

**Relation To Broader Scientific Literature:**

As I mentioned above, the paper has two main contributions.

(1) Using unit quaternions to represent rotations in protein backbone design.
(2) Using the ReFlow procedure to rectify the flow and get better generation with fewer sampling steps.

Regarding (1). This idea attempts to improve backbone design methods that represent residues as a translation+rotation. However, instead of representing rotations using 3x3 rotation matrices, the paper proposes to use unit quaternions, claiming they provide benefits in terms of stability. I consider this to be the main reason why (1) is a relevant contribution (but please correct me if I’m wrong). This being said, I feel section 3.4, which comments on stability and does a small empirical study for both representations, could be a lot more detailed. The paper “Exploring SO(3) logarithmic map: degeneracies and derivatives” by Nurlanov discussed in quite some detail the instabilities of the log map when dealing with 3x3 rotation matrices. In certain cases, a Taylor approximation can be used to avoid instabilities (eqs 9a and 9b in that paper). In other cases, due to numerical reasons, when the rotation angle is close to pi, things can only be determined up to a global sign (see paragraphs right before section 2.2). Which, if any, of these instabilities are being addressed by using unit quaternions? I think providing more details about this would be informative in understanding exactly which instabilities are being addressed, and how / why. Also, would be interested to have a clear discussion of which instabilities are left when working with unit quaternions, to fully understand how the two methods compare.

Regarding (2). I see this contribution as orthogonal to the method introduced, as most existing methods for protein backbone design could potentially benefit from ReFlow. However, to my knowledge this is the first paper to explore the use of ReFlow in this domain, showing promising results. However, doing ReFlow appears to affect the method’s performance (see “Other strengths and weaknesses” section below) quite a bit, not always positively.

**Theoretical Claims:**

The theoretical claims in the paper are extensions from the ReFlow [1] results (keeps correct marginals, reduces transport cost). I did not check the proof in detail.

[1] “Flow Straight and Fast: Learning to Generate and Transfer Data with Rectified Flow”, by Liu et al.

---

> ### Author Rebuttal · Authors · 2025-04-01
>
> Thanks for your positive feedback and constructive comments.
>
> Before resolving technical concerns,
>
> **1) Apology for reusing sentences in Proteina.** We sincerely apologize for reusing sentences from Proteina in Appendices B.5 and B.6 without proper citation. We read this paper when it was under review anonymously. We did not mention it in the related work because we focus on the tech route using frame-based representation + flow-based generative learning, while Proteina applies a very different route and did not release code at that time. In the appendix, originally, we had a sentence in the beginning of B.5, saying we follow existing evaluation methods, with the citations of Frameflow, FoldFlow, Genie2, and Proteina. However, it is accidentally commented when we adjusted the layout of Figure 6 and B.5. We thank the reviewers for pointing out our oversight.
>
> We take this matter seriously. **In the revised paper, we will 1) rewrite B.5 and B.6 completely in a different logic flow; 2) mention Proteina in the related work and introduce its tech route briefly; and 3) take the comparison with Proteina as our future work in the conclusion.**
>
> **2) Correction of long-chain results:** We found a bug in our script in the rebuttal phase and correct the results of ReQFlow on long-chain generation are in https://anonymous.4open.science/r/6342_ReQFlow_Rebuttal-62BB/01_revised_longchain.pdf. **Note that, the new results are lower than the wrong ones in the submission, but they are still significantly better than those of baselines. This correction does not affect our main claims and contributions.**
>
> Below we try to resolve your technical concerns one by one.
>
> **Q1. More metrics on long chain experiment**
>
> **A1:** As shown in https://anonymous.4open.science/r/6342_ReQFlow_Rebuttal-62BB/03_long_chain_metrics.pdf, both QFlow and ReQFlow outperform FrameFlow on designability, though their diversity and novelty metrics drop.
>
> We think this is reasonable because compared with FrameFlow, both QFlow and ReQFlow generate more designable proteins. There are more proteins used to compute novelty and diversity, which may lead to higher TM-scores because of the inherent trade-off among the metrics.
>
> In addition, currently, ReQFlow is not as good as larger models because 1) our training utilized only 23k samples, a small subset of the PDB, far fewer than Genie2 (590k), FoldFlow2 (160k), and RFDiffusion (>208k); 2) our model has much fewer parameters than RFDiffusion and FoldFlow2.
>
> Training a larger QFlow/ReQFlow model on a larger dataset can be one of our future work, but it is out of the scope of this work.
>
> **Q2. The reason for the stability of using quaternion.**
>
> **A2:**  When angle is close to $\pi$，the quaternion logarithm is simply `log(q) = (π/2)u`. This calculation is direct and numerically stable; there's no division by a small number.
>
> When angle is close to $0$, quaternion logarithm `log(q)` also faces potential instability. It is handled by clamping the divisor much simpler than the Taylor expansion.
>
> **Q3. A larger synthetic dataset for rectifying flow**
>
> **A3:** We use 17.6k synthetic data to train ReQFlow. However, using a larger dataset does not affect results significantly. As shown in **A1**, this is due to 1) a trade-off among the metrics and 2) the limited model size and data we used.
>
> ||Steps|Designability|Diversity|Novelty|
> |-|-|-|-|-|
> |Original: 7.7k|500|0.972|0.377|0.828|
> ||50|0.912|0.369|0.810|
> ||10| 0.676|0.337|0.760|
> |Larger: 17.6k|500|0.968|0.381|0.832|
> ||50|0.932|0.379|0.825|
> ||10|0.724|0.360|0.793|
>
> **Q4. The rationality and usefulness of ReFlow on designable proteins.**
>
> **A4 (Short answer):** Selecting designable proteins matters, but the self-distillation of QFlow on such proteins may suffer mode collapse. ReQFlow avoids this issue to some extent and maintains competitive results. Please refer to our response **A2 to Reviewer K7Q3** for detailed answer. https://openreview.net/forum?id=f375uEmYDf&noteId=40R3BVTqCr
>
> **Q5. The computation of the average of pairwise tm-scores**
>
> **A5:** The score is first done per length, and then averaged across lengths. We will revise our paper to make it clear.
>
> **Q6. Setting of baselines.**
>
> **A6:** All the baselines are evaluated using the default settings in their repos. For Genie2, we use the default setting (noise scale=0.6). We will revise our paper to make it clear.
>
> **Q7. Explain the power of exponential step scheduler.**
>
> **A7 (Short answer):** **current method models velocity field backward**, making the prediction at the starting point difficult and inaccurate. Exponential scheduler allows models to approach the ending point with few steps, avoiding severe error accumulations near the starting point. Please refer to our response **A4 to Reviewer K7Q3** for more details. https://openreview.net/forum?id=f375uEmYDf&noteId=40R3BVTqCr
>
> In summary, we hope our responses can resolve your concerns and enhance your confidence to further support our work.

---

### Official Review · Reviewer_L7Bp · 2025-03-24

**Overall Recommendation:** 4

**Summary:**

This paper introduces a new flow matching method for unconditional protein backbone generation, based on quaternion representations and rectified flows.
More specifically in this work, the rotational part of the backbone residues is represented as unit quaternion, instead of an SO(3) matrix which is the canonical choice in most previous work. It is demonstrated that this choice improves the numerical stability of calculations, leading to improved backbone quality as well as a speed up of the inference process.
Secondly the “reflow” method from rectified flows is applied to pretrained models, leading to straighter flow trajectories and further improving the quality of sampled backbones.

## Update after rebuttal

I thank the authors for their detailed response and the additional
experiments. The new results, in my view, enhance the clarity of the
paper and further highlight its core contribution: a robust and
numerically stable framework for protein backbone design. I will thus
increase my score and recommend to accept the paper.

**Claims And Evidence:**

The claims about increased numerical stability and higher quality backbones are supported by sensible experiments and results.
The application of the reflow procedure on a curated set of training samples significantly increases designability while trading off diversity and novelty, as is expected.
The comparison of ReQFlow and ReFrameFlow indicates an increase in designability of approximately 8% and a 25% speedup of ReQFlow compared to Frameflow. Unfortunately no detailed summary of training hyperparameters is provided, which would allow for a better understanding of the differences between the two models. The time threshold of 0.6 for the auxiliary loss in eq. (50) and (51) indicates that the hyperparameters are not the same as in FrameFlow.

**Essential References Not Discussed:**

All important references are included.

**Experimental Designs Or Analyses:**

The experimental design choices seem sensible and are in accordance with common choices in the field of unconditional protein backbone generation. In particular, the definition of the metrics designability, diversity and novelty as well as sampling steps and choices of backbone lengths to generate are the same as for multiple other baselines.

**Methods And Evaluation Criteria:**

The comparison of numerical stability of matrix and quaternion implementations in terms of a round-trip error is appropriate, given that similar operations are performed when calculating the geodesics on SO(3).
The benchmark datasets SCOPe and the curated version of PDB are widely adopted in the field.
The performed experiments cover most interesting benchmarks for unconditional protein backbone generation models, including evaluations of designability, diversity, novelty and secondary structure content in various settings.

**Other Comments Or Suggestions:**

For the computation of novelty, Fold-seek was used. Fold-seek has an issue where the TM-Score is provided in the wrong column of the output. The command provided in the supplementary suggests that this error might affect the novelty results reported in the tables? (see https://github.com/steineggerlab/foldseek/issues/323)

**Other Strengths And Weaknesses:**

Weakness:
The paper claims that the IGSO(3) prior corresponds to uniformly sampling rotation axis and rotation angle, which is not the case (see e.g. discussion in Leach et al. and Yim et al.). Crucially many other baselines use the IGSO(3) prior during training and a uniform prior on SO(3) during inference. Clarification on what is implemented for QFlow would be desirable.

Leach, Adam, et al. "Denoising diffusion probabilistic models on so (3) for rotational alignment." (2022).
Yim, Jason, et al. "Fast protein backbone generation with se (3) flow matching." arXiv preprint arXiv:2310.05297 (2023).

**Questions For Authors:**

1.	Could you please provide a more detailed list of training hyperparameters and indicate if and how the QFlow models are different from FrameFlow? In the case of equal hyperparameters could you explain how the increased numerical stability for rotation angles close to π can lead to such a significant increase in designability?
2.	The reported 25% speedup over FrameFlow seems quite large. Could you provide more details on the specific computational bottlenecks that QFlow optimizes and the corresponding speedup for each of these individual components?
3.	For the ablation results in Table 4 I would like to see results on more than one checkpoint to evaluate the statistical significance of the reported metrics. This would also allow for a better comparison between the models.
I am willing to raise my score if the above points are addressed in the rebuttal.

**Relation To Broader Scientific Literature:**

The proposed method directly builds on established flow matching models for protein backbone generation in particular FrameFlow. The usage of reflow is a direct application of the ideas in Liu et al. The idea of using quaternion representations and SLERP in exponential format for handling the rotations of residue frames is novel and could be readily applied also to many other models in the field which work with frame representations for backbone residues.

Liu, Xingchao, Chengyue Gong, and Qiang Liu. "Flow straight and fast: Learning to generate and transfer data with rectified flow." arXiv preprint arXiv:2209.03003 (2022).

**Theoretical Claims:**

The theoretical claims of this paper are mainly concerned with repeating the proofs of Liu et al. for the case of quaternion algebra and seem correct.

Liu, Xingchao, Chengyue Gong, and Qiang Liu. "Flow straight and fast: Learning to generate and transfer data with rectified flow." arXiv preprint arXiv:2209.03003 (2022).

---

> ### Author Rebuttal · Authors · 2025-04-01
>
> Thanks for your positive and constructive comments.
>
> **Correction of long-chain results:** Before resolving your concerns, we would like to report that we found a bug in our script in the rebuttal phase and correct the results of ReQFlow on long-chain generation in the anonymous link https://anonymous.4open.science/r/6342_ReQFlow_Rebuttal-62BB/01_revised_longchain.pdf. **Note that, the new results are lower than the wrong ones in the submission, but they are still significantly better than those of baselines, and thus, do not affect our main claims and contributions.**
>
> Below we try to resolve your concerns one by one.
>
> **Q1. Hyperparameters of QFlow/ReQFlow**
>
> **A1.** We adopt the the same hyperparameter settings as FrameFlow for a fair comparison, and the key parameters are shown below:
> |Hyperparam.|Value|
> |-|-|
> |aux\_loss\_t\_pass (time threshold)|PDB=0.5, SCOPe=0.25|
> |aux\_loss\_weight|1.0|
> |batch size|128|
> |max\_num\_res\_squared|PDB=1000000, SCOPe=500000|
> |max epochs|1000|
> |learning rate|0.0001|
> |interpolant\_min\_t|0.01|
>
> The batch size of FrameFlow is originally set to 128 for PDB and 100 for SCOPe. We retrained FrameFlow on SCOPe in our work and set the batch size to 128, the same as QFlow/ReQFlow. This doesn't influence the comparison's effectiveness and fairness.
>
> In addition, the value "0.6" in Eqs. (50, 51) does not represent a time threshold but rather an inter-atomic distance threshold, with the unit nanometers (nm).
>
> **Q2. Clarification on SO(3) prior**
>
> **A2.** Following FrameFlow, we use an IGSO(3) prior for training and a uniform prior on SO(3) for inference. We apologize for the mistake on omitting the density. The text on the right side of lines 149-150 actually means "uniformly sampling ... with the following density:"
> $$
> f(\omega) = \frac{1 - \cos\omega}{\pi} \sum_{l=0}^{\infty} (2l + 1) e^{-l(l+1)\epsilon^2} \frac{\sin\left(\left(l + \frac{1}{2}\right)\omega\right)}{\sin(\omega/2)}.
> $$
> This description is the same as Leach, Adam, et al.'s definition. We will clarify this point in the revised version.
>
> **Q3: Issue on Fold-seek whether affect novelty results.**
>
> **A3:** Thank you for pointing out this known issue (Foldseek Issue #323). We were aware of this issue where the `evalue` column reports the TM-score in TM-align mode (`--alignment-type 1`). To avoid this issue, as shown in our supplementary command, we requested the TM-score using the `--format-output ... alntmscore, ...` mode. Therefore, we utilized the correct `alntmscore` column for our analysis, ensuring our reported novelty results based on TM-scores are accurate and unaffected by this issue.
>
> **Q4. Why numerical stability on large ablges (close to $\pi$) increases designability.**
>
> **A4.** As mentioned in Section 3.4, lines 258–260, during the inference, the probability of suffering at least one large angle ($\geq \pi - 10^{-2}$) per protein is 0.59 for PDB and 0.34 for SCOPe. Meanwhile, Figure 2(a) illustrates that when the angle is close to $\pi$ ($\geq \pi - 10^{-2}$), the matrix’s mean round-trip error increases dramatically. Because this round-trip step (i.e., conversions between rotation angles and rotation matrices/quaternions) is a critical and high-frequent operation of interpolation during inference, the round-trip errors propagate and lead to significant orientation deviations in residues during inference.
>
> **Q5. Why QFlow/ReQFlow speedup over FrameFlow**
>
> **A5.** We had an in-depth analysis of the code, and the speed improvements can be attributed to the following 3 components.
>
> **1) Fewer float-point operations:** When describing a 3D rotation, the matrix multiplication (27 mul., 18 add.) is intrinsically more computationally expensive than a quaternion multiplication (16 mul., 12 add.).
>
> **2) Fewer matrix-vector multiplications:** For each residue, matrix-based interpolation performs 3 matrix multiplication (i.e., computing relative rotation matrix, implementing the Rodrigues formula, and applying one rotation matrix to the initial matrix) while quaternion-based performs 2 quaternion multiplications (i.e., two Hamilton products).
>
> **3) Cheaper nonlinear operations:** The matrix-based log/exp maps require to handle numerical issues (although still failed on angles $\geq \pi - 10^{-2}$ ), e.g., using truncated Taylor approximation. In contrast, quaternions rely on simple mathematical operations, such as acos, sin, cos, and normalization (sqrt/division).
>
> **Q6. Ablation results on more checkpoints**
>
> **A6.** For the ablation results, we use 5 checkpoints to evaluate their statistical significance, demonstrating their consistency and stability.
> |Exponential Scheduler|Flow Rectification|Data Filtering|500|50|10|
> |-|-|-|-|-|-|
> |x|x|x|0.143±0.079|0.047±0.030|0.002±0.002|
> |✓|x|x|0.910±0.029|0.795±0.051|0.309±0.058|
> |✓|✓|x|0.612±0.084|0.519±0.154|0.385±0.136|
> |✓|✓|✓|0.969±0.027|0.932±0.022|0.698±0.041|
>
> We hope the above answers help resolve your concerns and enhance your confidence to support our work.

---

> > ### Comment · Reviewer_L7Bp · 2025-04-04
> >
> > Answer to rebuttal:
> >
> > I thank the authors for their response. I would like to follow up on some of the points:
> >
> > Q3: Looking at the command in the Appendix and also comparing the reported novelty with values for FrameFlow computed using the numbers from the evalue column I still think that the reported values might be incorrect. I would encourage the authors to have another look at this issue and take the values directly from the evalue column. This should actually make the results better.
> >
> > Q4: I acknowledge the fact that the improved numerical precision can lead to higher quality backbones. However, I still have the concern that the reported increase in designability can not only be attributed to the usage of quaternions over rotation matrices. In my experience, when training FrameFlow, the designabilities between checkpoints show large fluctuations (up to 10%) even if the loss has already converged. It would thus be good to know how the checkpoints were selected and what measures were taken to ensure that the above-mentioned effect did not distort the results.
> >
> > Q5: The discussed differences between matrix and quaternion algebra show that the use of quaternions is beneficial for computational performance. I am however still skeptical about the magnitude of the acceleration. I would suspect that looking at the full pipeline the neural network is a much bigger bottleneck than the update of the frames. Do you have numbers for the relative compute time of the forward pass of the neural network compared to the frame updates?
> >
> > Q6: Thank you for providing errors for the values in table 3, I think that they are very helpful to judge the results. I my original review I however referred to table 4, since I think errors are especially important for the comparison between the FrameFlow and QFlow models (see also the point on Q4). I think it would be beneficial if these were included in the final version of the paper.
> >
> > For now, I keep my current rating.

---

> > > ### Author Response · Authors · 2025-04-08
> > >
> > > Thank you for your insightful feedback. In the past few days, we added more experiments for resolving your remaining concerns.
> > >
> > > **Q3. The correction of novelty computation**
> > >
> > > **A3:** Thank you for your suggestion. We take the values directly from the E-value column to update our novelty computations. The revised Tables 2 and 4 are shown in https://anonymous.4open.science/r/6342_ReQFlow_Rebuttal_Second_Phase-2F4E/revised_table_2_4.pdf.
> > > On PDB, QFlow/ReQFlow shows a trade-off between designability and novelty, while on SCOPE, QFlow/ReQFlow achieves higher designability with competitive novelty scores.
> > >
> > > **Q5. Comparison for QFlow and FrameFlow on speed.**
> > >
> > > **A5:** Thanks for your comment. We analyzed the runtime of QFlow and FrameFlow and found an implementation discrepancy: In the `interpolant.sample` function, when reconstructing the protein frame trajectory into atomic coordinates, our QFlow implementation only considers the first and last proteins, whereas FrameFlow reconstructs all intermediate steps.
> > >
> > > For a fair comparison, we reconstruct the first and last proteins for both methods and record their runtime (second) on generating a protein of length 300 in the PDB experiment and length 128 in the SCOPe experiment:
> > >
> > > |Datasets|Methods|Steps|Model Prediction|Rotation Update|Translation Update|Total Time|
> > > |-|-|-|-|-|-|-|
> > > |PDB|FrameFlow|500|16.308±0.093|0.608±0.005|0.033±0.000|17.053±0.099|
> > > |||50|1.609±0.013|0.059±0.001|0.003±0.000|1.727±0.014|
> > > |||20|0.635±0.008|0.024±0.001|0.001±0.000|0.713±0.010|
> > > ||QFlow|500|16.732±0.089|0.492±0.004|0.036±0.000|17.370±0.111|
> > > |||50|1.670±0.003|0.048±0.000|0.003±0.000|1.776±0.004|
> > > |||20|0.653±0.001|0.019±0.000|0.001±0.000|0.726±0.002|
> > > |SCOPe|FrameFlow|500|11.947±0.125|0.601±0.003|0.033±0.000|12.688±0.124|
> > > |||50|1.166±0.013|0.059±0.001|0.003±0.000|1.275±0.016|
> > > |||20|0.471±0.002|0.025±0.000|0.001±0.000|0.539±0.003|
> > > ||QFlow|500|11.994±0.037|0.483±0.003|0.034±0.000|12.602±0.040|
> > > |||50|1.166±0.015|0.048±0.001|0.003±0.000|1.262±0.021|
> > > |||20|0.466±0.002|0.019±0.000|0.001±0.000|0.528±0.002|
> > >
> > > Your intuition is correct: the neural network feedforward computation is the main computational bottleneck. However, we believe this new result does not diminish our core contributions --- 1) The quaternion operations are 15~20% faster than rotation matrix-based operations (see the Rotation Update column), and 2) applying quaternions indeed leads to better numerical stability and improves efficiency by reducing inference steps while maintaining high designability.
> > >
> > > We will add the above efficiency analysis in the revised paper, and we have update the runtime results, see https://anonymous.4open.science/r/6342_ReQFlow_Rebuttal_Second_Phase-2F4E/revised_table_2_4.pdf.
> > >
> > >
> > > **Q4 & Q6. Checkpoint selection + Verify the contribution of quaternion operation.**
> > >
> > > **A4 & A6:** **Checkpoint selection strategy.** For each method, after observing loss convergence, we select checkpoints based on the metrics of the generated protein validation set. We choose the checkpoint where the `ca_ca_valid_percent` > 0.99 and the proportions of secondary structures are closest to the dataset's average values. For example, when selecting the checkpoint of QFlow on PDB, the generated protein validation set has a `ca_ca_valid_percent` = 0.996, a `helix_percent` = 0.400, and a `strand_percent` = 0.283.
> > >
> > > **Results in Table 4 achieved by different checkpoints.** To demonstrate that the improvement is attributed to quaternion operations, we show the performance below using five checkpoints for each model (Computing the novelty score is time-consuming while the rebuttal time is limited, we report one checkpoint's novelty in https://anonymous.4open.science/r/6342_ReQFlow_Rebuttal_Second_Phase-2F4E/revised_table_2_4.pdf and report multi-checkpoint's designability and diversity here). The result of each checkpoint is in https://anonymous.4open.science/r/6342_ReQFlow_Rebuttal_Second_Phase-2F4E/statistical_significance.pdf.
> > >
> > > ||Step|Fraction|scRMSD|Diversity|
> > > |-|-|-|-|-|
> > > |FrameFlow|500|0.851±0.016|1.437±0.035|0.392±0.007|
> > > ||50|0.811±0.017|1.566±0.053|0.378±0.008|
> > > ||20|0.708±0.023|1.966±0.089|0.370±0.005|
> > > |QFlow|500|0.893±0.022|1.288±0.082|0.392±0.005|
> > > ||50|0.854±0.019|1.441±0.048|0.380±0.005|
> > > ||20|0.762±0.015|1.766±0.051|0.372±0.004|
> > > |ReFrameFlow|500|0.924±0.007|1.213±0.031|0.407±0.004|
> > > ||50|0.906±0.006|1.268±0.004|0.407±0.001|
> > > ||20|0.884±0.009|1.399±0.032|0.405±0.004|
> > > |ReQFlow|500|0.947±0.007|1.131±0.025|0.406±0.003|
> > > ||50|0.922±0.007|1.189±0.024|0.411±0.002|
> > > ||20|0.910±0.012|1.282±0.038|0.405±0.001|
> > >
> > > The results show that the designability improvement by using quaternion is stable and robust. These results, including novelty scores, will be included in the final version of the paper.
> > >
> > > **We hope that the above responses can resolve your remaining concerns completely and enhance your confidence to raise your score. We would appreciate it if you can further support our work in the following discussion and decision phases.**

---

### Decision · Program_Chairs · 2025-05-01

**Decision:**

Accept (poster)

**Comment:**

This submission proposes ReQFlow, a new flow matching framework for protein backbone generation. ReQFlow uses unit quaternions to represent 3D rotations, which improves numerical stability and enables faster computation. It also incorporates rectified flows to enhance sampling quality and efficiency.

The reviewers have recognized the significance of the paper’s contributions and acknowledged the strong empirical results. Reviewer TTp8 noted that a couple of paragraphs describing the evaluation metrics were copied from prior work. I agree with the reviewer that this issue is not substantial enough to warrant rejection, but I strongly encourage the authors to revise this in the final version or give propert citation to the relevant work.

As pointed out by reviewers TTp8 and K7Q3, the authors should soften their claims from "state-of-the-art" to "on-par" and include a discussion or visualization of the trade-offs between designability, novelty, and diversity.

Overall, the strengths of this submission outweigh its weaknesses, and I am happy to recommend acceptance.